# Progression of herpesvirus infection remodels mitochondrial organization and metabolism

**Simon Leclerc[1], Alka Gupta[1], Visa Ruokolainen[1], Jian-Hua Chen[2], Kari Kunnas[1], Axel A. Ekman[1,2], Henri Niskanen[3], Ilya Belevich[4], Helena Vihinen[4], Paula Turkki[5] Ana J. Perez-Berna[6], Sergey Kapishnikov[7], Elina Mäntylä[5], Maria Harkiolaki[8], Eric Dufour[5], Vesa Hytönen[5,9], Eva Pereiro[6], Tony McEnroe[7], Kenneth Fahy[7], Minna U. Kaikkonen[3], Eija Jokitalo[4], Carolyn A. Larabell[2,10], Venera Weinhardt[11], Salla Mattola[1], Vesa Aho[1]☯*, Maija Vihinen-Ranta ☉[1]☯***

1 Department of Biological and Environmental Science and Nanoscience Center, University of Jyvaskyla, Jyvaskyla, Finland, 2 Molecular Biophysics and Integrated Bioimaging Division, Lawrence Berkeley National Laboratory, Berkeley, California, United States of America, 3 A.I. Virtanen Institute for Molecular Sciences, University of Eastern Finland, Kuopio, Finland, 4 Electron Microscopy Unit, Institute of Biotechnology, Helsinki Institute of Life Science, University of Helsinki, Finland, 5 BioMediTech, Faculty of Medicine and Health Technology, Tampere University, Tampere, Finland, 6 MISTRAL Beamline-Experiments Division, ALBA Synchrotron Light Source, Cerdanyola del Valles, Barcelona, Spain, 7 SiriusXT Limited, Dublin, Ireland, 8 Diamond Light Source, Harwell Science and Innovation Campus, Didcot, UK; Division of Structural Biology, The Henry Wellcome Building for Genomic Medicine, Roosevelt Drive, Oxford, United Kingdom, 9 Fimlab laboratories, Tampere, Finland, 10 Department of Anatomy, University of California San Francisco, San Francisco, California, United States of America, 11 Centre for Organismal Studies, University of Heidelberg, Heidelberg, Germany

☯ These authors contributed equally to this work.
* vesa.p.aho@jyu.fi (VA); maija.vihinen-ranta@jyu.fi (MV-R)

## Abstract

Viruses target mitochondria to promote their replication, and infection-induced stress during the progression of infection leads to the regulation of antiviral defenses and mitochondrial metabolism which are opposed by counteracting viral factors. The precise structural and functional changes that underlie how mitochondria react to the infection remain largely unclear. Here we show extensive transcriptional remodeling of protein-encoding host genes involved in the respiratory chain, apoptosis, and structural organization of mitochondria as herpes simplex virus type 1 lytic infection proceeds from early to late stages of infection. High-resolution microscopy and interaction analyses unveiled infection-induced emergence of rough, thin, and elongated mitochondria relocalized to the perinuclear area, a significant increase in the number and clustering of endoplasmic reticulum-mitochondria contact sites, and thickening and shortening of mitochondrial cristae. Finally, metabolic analyses demonstrated that reactivation of ATP production is accompanied by increased mitochondrial $Ca^{2+}$ content and proton leakage as the infection proceeds. Overall, the significant structural and functional changes in the mitochondria triggered by the viral invasion are tightly connected to the progression of the virus infection.

**Data Availability Statement:** The authors declare that the GRO-Seq data were deposited into the Gene Expression Omnibus database under accession number GSE243613 and are available at

https://www.ncbi.nlm.nih.gov/geo/query/acc.cgi?acc=GSE243613. All other relevant data are in the manuscript and its supporting information files.

**Funding:** This work was financed by the Jane and Aatos Erkko Foundation (MVR); Academy of Finland under award number 330896 (MVR) and 332615 (EM); European Union's Horizon 2020 research and innovation program under grant agreement No 101017116, project Compact Cell-Imaging Device (CoCID; EP, TM, KF, VW, MVR); with the support of Biocentre Finland and Tampere Virus Production Facility (ED). This project benefited from access to ALBA and has been supported by iNEXT-Discovery, project number 871037, funded by the Horizon 2020 program of the European Commission (EP). This study was funded by ALBA Synchrotron standard proposals 2021095277, 2022025597, and 2022086951 (EP). The funders had no role in study design, data collection and analysis, decision to publish, or preparation of the manuscript.

**Competing interests:** The authors have declared that no competing interests exist.

## Author summary

Herpesviruses not only cause significant diseases but are also promising candidates for oncolytic therapy. The HSV-1 infection depends on the nuclear DNA replication, transcription machinery, and mitochondrial metabolism of the host cell. Late in lytic infection, HSV-1 induces major structural changes in nuclear structures, including host chromatin, and mitochondria. This study investigated time-dependent mitochondrial changes as HSV-1 infection proceeds from early to late infection. We show that infection leads to significant transcriptional modification of genes encoding proteins involved in the mitochondrial network, such as the respiratory chain, apoptosis, and the structural organization of mitochondria. Our findings indicate that infection leads to significant alterations in mitochondrial structure and function, including changes in mitochondrial morphology and distribution, thickening and shortening of cristae, an increase in the number and area of contact sites between mitochondria and the endoplasmic reticulum, as well as a rise in mitochondrial calcium ion content and proton leak.

## Introduction

Mitochondria are double membrane-bound, ATP-producing powerhouses of the cells comprising approximately 1,000 proteins [1], and mitochondrial metabolism produces precursors for biosynthetic pathways including nucleotides, lipids, and amino acids [2–4]. The mitochondria also have a role in the innate immune system including the ability to respond to various types of internal and external stressors. Both the internal stressors such as genetic, metabolic, and biochemical factors, as well as external stressors such as environmental agents lead to dynamic functional and morphological alterations of the mitochondria [5,6].

One of the external stressors that affect the mitochondria is a viral infection [7]. Production of viral proteins is followed by stimulation of mitochondria-mediated immune signaling and cell death pathways [8,9]. The mitochondria have an important role in the mediation of programmed cell death, and viruses target this machinery either to ensure the viability of cells for the viral replication or to destroy them for more efficient spreading of progeny viruses [10]. Many mitochondrial changes, including those related to apoptosis, are triggered by calcium influx from the endoplasmic reticulum (ER). The mitochondria and ER are often physically located next to each other and connected via specific contact sites. The formation and maintenance of the contact sites are mediated by tethering proteins, such as VAPB and PTPIP51 [11]. Several viruses have been shown to target the calcium homeostasis between the ER and mitochondria either to promote or inhibit apoptosis [12–14]. Besides using various mechanisms to evade cellular antiviral responses, the progression of viral replication relies on the reprogramming of the host cell mitochondrial metabolism. Severe acute respiratory syndrome coronavirus 2 (SARS-CoV-2) and human immunodeficiency virus (HIV) infections induce disturbance of mitochondrial homeostasis, which leads to increased glycolysis and production of metabolites useful in the synthesis of lipids, nucleotides, and proteins needed during the formation of viral particles [15–18]. Herpes simplex virus type 1 (HSV-1) supports the host cell mitochondrial oxidative and biosynthetic metabolism by inducing replenishment of tricarboxylic acid cycle intermediates [19]. Another herpesvirus, human cytomegalovirus (HCMV), activates mitochondrial energy production by increasing glycolysis [19,20].

The alpha-herpesvirus HSV-1 is an enveloped double-stranded DNA virus that hijacks the cellular metabolism including mitochondrial biosynthetic pathways to produce precursors required for replication [21]. The regulation of apoptosis to prevent the premature death of the

host cells is essential for the envelopment and nonlytic cellular egress of HSV-1 [9,22,23]. The viral proteins ICP4 and Us3 suppress apoptosis by inhibiting caspase activation and inactivating proapoptotic proteins, and ICP27 protein counteracts caspase 1-dependent cell death [24–26]. The HSV-1 infection has been shown to cause relocalization of mitochondria towards the nuclei as well as elongation of mitochondria [27,28]. In the absence of infection, elongation can be induced by starvation and inflammatory response [29]. The elongation can protect mitochondria from autophagy and lead to an increase in the surface area of cristae to enhance ATP production [30].

Here, we analyze how the progression of HSV-1 infection influences mitochondrial gene transcription, ultrastructure, and organization, and how it contributes to mitochondrial function. We use various microscopical and biochemical tools uniquely suited for observing detailed changes in mitochondrial organization and function of HSV-1-infected cells. Our approach provides characterization and quantification metrics for the mitochondrial protein transcription, mitochondrial structures including cristae, and energy metabolism at the early and late stages of infection. We demonstrate that late infection-induced significant changes in the mitochondrial protein gene transcription are associated with dramatic changes in the mitochondrial ultrastructure and distribution, ER contacts, and function.

## Results

### The transcription of mitochondria-associated proteins is modified in HSV-1 infection

To explore virus-induced changes in the host cell transcription, we employed a global run-on sequencing (GRO-seq) analysis of nuclear-encoded genes in green monkey kidney epithelial (Vero) cells. The clustering of genes in the same pathways (based on the STRING database (https://string-db.org/) of predicted functional protein-protein interactions) revealed that the infection-induced changes in the gene transcription were linked to proteins that contribute to mitochondrial functions such as electron transport, immunity, and apoptosis (Fig 1A and 1B). A substantial increase in functional interactions and regulation of transcription was detected when infection proceeded from 4 to 8 hpi. (Figs 1A, 1B, S1A and S1B). In general, the transcription of 43 and 59 nuclear-encoded mitochondrial genes (hereinafter simply referred to as mitochondrial genes) was upregulated, while 22 and 102 were downregulated at 4 and 8 hpi, respectively, in comparison to the noninfected cells (S1B Fig). To assess the effect of interferon deficiency of Vero cells [31] in the transcription of mitochondria-associated proteins, we analyzed interferon-producing Hela cells [32] at 4 and 8 hpi. The GRO-seq data revealed that the general cellular transcription pattern in HeLa cells was similar to the Vero cells, however, with some cell line-specific variations (S2A and S2B Fig). The progression of infection in Vero cells reduced the transcription of genes involved in energy metabolism. One of the key components of the electron transport chain, the first enzyme of the respiratory chain, is complex I (NADH-ubiquinone oxidoreductase) [33]. Two and nine of the genes encoding proteins involved in this 45-protein complex, consisting mostly of nuclear-encoded mitochondrial proteins (NDUF subunits), were downregulated at 4 and 8 hpi, respectively. As expected, activation of mitochondrial antiviral defenses led to the regulation of genes associated with apoptosis and immune response (Figs 1A, 1B and S1B). For example, the apoptosis-inducing protein BCL2L11 (also known as BIM), a member of the BCL2 family regulating the intrinsic, mitochondrion-dependent apoptosis [34], was upregulated both at 4 and 8 hpi. Transcription of the gene encoding histone deacetylase Sirtuin 1 (Sirt1), a multifunctional protein regulating apoptosis by controlling the cellular distribution of p53 [35], also increased in infected cells (Fig 1A and 1B and S1 Table). In contrast, transcription of the gene encoding tumor necrosis

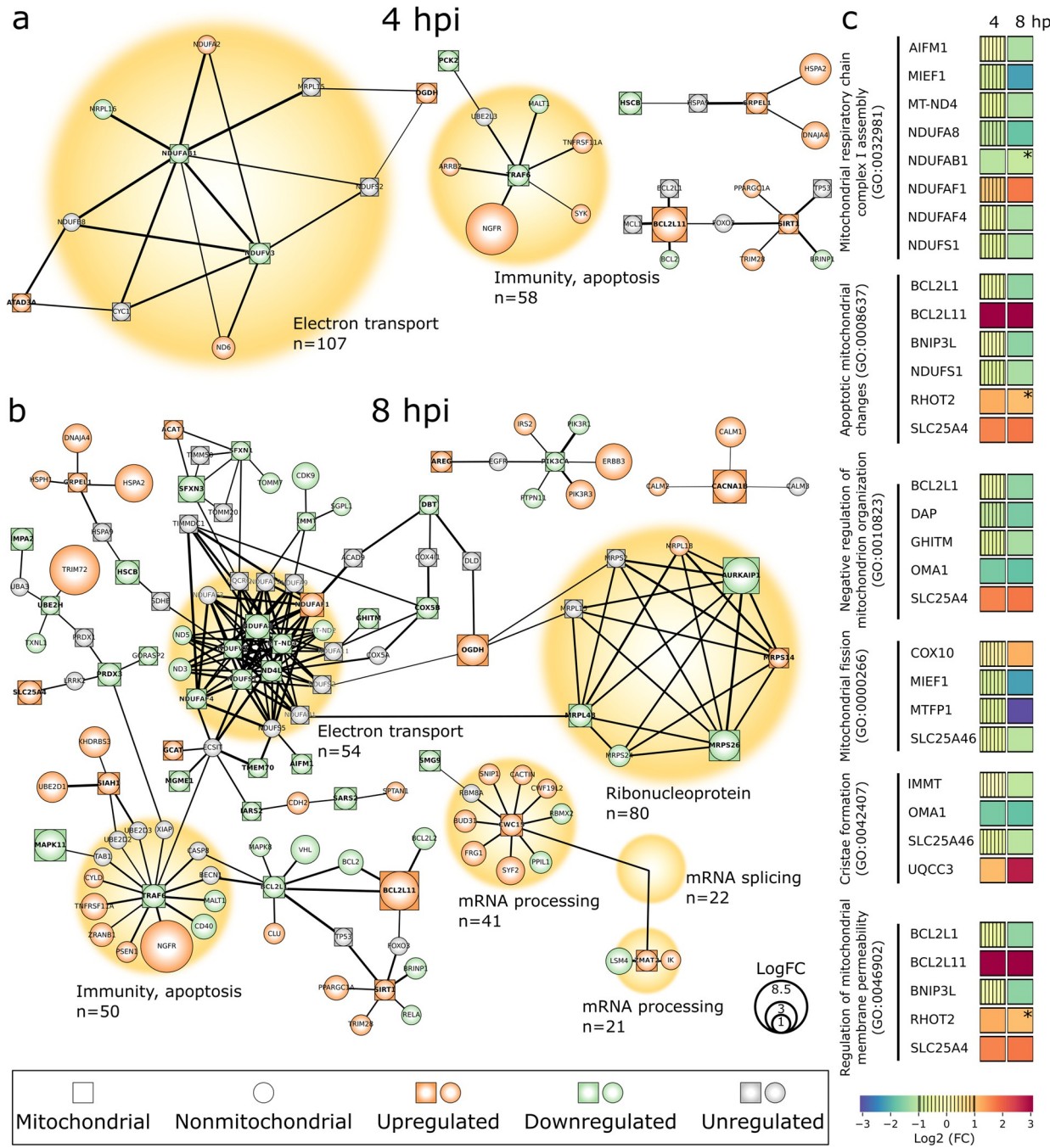

**Fig 1. HSV-1 infection alters the host transcriptome.** Global run-on sequencing (GRO-seq) analysis of nascent RNA levels of mitochondrial and mitochondria-associated proteins in infected Vero cells at **(a)** 4 and **(b)** 8 hpi. The genes encoding for mitochondrial proteins (square-shaped nodes, bolded titles) and nonmitochondrial cellular interactor proteins (round-shaped nodes) are shown. The upregulated (orange) or downregulated (green) transcripts in response to infection are visible together with unregulated interacting transcripts (grey). The node size correlates with a logarithmic fold change (logFC) of regulation and the thickness of black lines between nodes is proportional to the interaction in the STRING database (https://string-db.org/). The main functions of the proteins are denoted with yellow circles with their size proportional to the number of interactors. **(c)** GO term classification for mitochondrial processes in the infected cells at 4 and 8 hpi. The color bar indicates upregulation (orange-red) or downregulation (green-blue), and low change of gene transcription (vertical stripes). The non-significant enrichment is also shown (*).

factor receptor-associated factor 6 (TRAF6), a member of the TRAF family of proteins mediating signaling pathways that regulate inflammatory signaling, was downregulated both at 4 and 8 hpi. By suppressing cell death complex assembly, TRAF6 can inhibit tumor necrosis factor α (TNF-α)-induced apoptosis and necroptosis [36]. Moreover, the upregulated transcription of ATAD3A of membrane protein AAA domain-containing protein 3 member A (ATAD3A) at 4 hpi, involved in the removal of damaged mtDNA [37], is in line with the HSV-1 infection-induced disintegration of the mtDNA [27,38]. Finally, the infection increased the transcription of mitochondrial calcium voltage-gated channel subunit alpha1 B (CACNA1B) at 4 and 8 hpi.

To further assess the role of mitochondrial genes differentially expressed during the infection, we performed the gene ontology analysis (GO) according to the PANTHER classification system for the GO Biological process (http://www.pantherdb.org). The enriched GO terms associated with the regulation of mitochondrial organization and function were mitochondrial respiratory chain complex I assembly (GO:0032981), apoptotic mitochondrial changes (GO:0008637), negative regulation of mitochondrion organization (GO:0010823), mitochondrial fission (GO:0000266), cristae formation (GO:0042407), and regulation of mitochondrial membrane permeability (GO:0046902) (Fig 1C and S2 Table). The transcription of genes encoding proteins in the mitochondrial respiratory chain was mostly downregulated at 8 hpi. The downregulated genes corresponding to complex I proteins included NADH dehydrogenase subunit 4 (MT-ND4) and several nuclear-encoded NDUF subunits, and only the gene encoding NDUFAF1 was upregulated. In general, infection-induced stress results in cellular and viral pro- and antiapoptotic responses. Viral factors aim to enhance the envelopment and cellular exit of HSV-1 virions by preventing premature cell death [39,40]. On the other hand, we found that cellular proapoptotic processes included the upregulation of gene encoding apoptosis-inducing protein BCL2L11 and an integral outer mitochondrial membrane protein SLC25A46 (also known as adenine nucleotide translocator 1, ANT1). SLC25A46 is a multifunctional protein involved in mitochondrial ultrastructure alteration including the formation of elongated mitochondria, cristae maintenance, the enhancement of mitochondrial respiration/glycolysis, reactive oxygen species production, oxidative stress, and function of mitochondrial/ER contact that facilitates lipid transfer [41,42]. In infected cells, the upregulation of SLC25A46 was also involved in the negative regulation of mitochondrion organization and regulation of mitochondrial membrane permeability affecting proton gradient (Fig 1C and S2 Table). Moreover, the gene encoding ubiquinol-cytochrome C reductase complex assembly factor 3 (UQCC3 also known as C11orf83), with a role in cristae formation [43], was upregulated at 8 hpi. The downregulation of mitochondrial inner membrane protease (OMA1) is involved in the negative regulation of mitochondrion organization and cristae formation [44]. Activation of OMA1 during apoptosis and cellular stress leads to cleavage of another membrane protein, optical nerve atrophy 1 (OPA1), and results in the remodeling of cristae, the release of cytochrome c, and fragmentation of mitochondria [44,45]. Also, the genes encoding growth hormone-inducible transmembrane protein (GHITM, also known as TMBIM5 or MICS1), mitochondrial fission process 1 (MTFP1), and the inner membrane mitochondrial protein (IMMT, also known as Mic60, HMP), with roles in mitochondria morphology dynamics, fission, cristae organization, and contact sites [46–49], were downregulated at 8 hpi.

Altogether, we conclude that the progression of infection triggers timely responses of mitochondrial functions through regulation of transcription. The up- and downregulation of nuclear-encoded mitochondrial genes most likely reflect the balance between the cellular counteracts against the viral infection and the viral attempts to change cell functions to favor viral replication.

## Infection triggers changes in mitochondrial size and shape

To create a comprehensive view of virus-induced changes in the 3D structure and distribution of mitochondria, we analyzed suspended adherent mouse embryonic fibroblast (MEF) cells in glass capillaries and adherent MEF cells grown on electron microscopy (EM) grids using cryo soft X-ray tomography (SXT). SXT imaging is a non-invasive method for high-contrast imaging of the internal structure of intact frozen cells[48]. This method allows label-free imaging of carbon- and nitrogen-containing sub-cellular organelles. The image contrast arises from the attenuation of soft X-rays when they interact with the cellular internal structures and is dependent on the concentration and composition of biomolecules, also named linear absorption coefficient (LAC). Specifically, densely packed proteins or lipids attenuate strongly [49–51]. The SXT imaging in capillaries allows the 3D full-rotation tomography of the whole cells, whereas imaging cells on grids has reduced field of view. However, adherent MEF cells have to be detached before placing them into capillaries, whereas SXT imaging on grids provides the most native ultra-structure of the cell (S4 Table). Cryo-SXT images of suspended cells showed the development of elongated mitochondria at 4 hpi and increasingly at 8 hpi (Fig 2A and 2B and S1 Movie). The LAC values from the isotropic reconstructed 3D tomograms were measured for each voxel of the segmented 3D mitochondria and showed a significant increase at 4 and 8 hpi compared to the noninfected cells. This suggests that the infection led to an increased density of mitochondrial biomolecules (Fig 2C). Cryo-SXT data analysis also showed that the number of mitochondria increased at 4 hpi, and then decreased as the infection progressed to 8 hpi, suggesting that the fragmentation of mitochondria at early infection was followed by their fusion at late stages of infection (Fig 2D). However, the average volume of mitochondria increased as infection proceeded which argues against the disintegration (Fig 2E). As the segmentation of SXT data was challenged by artificial fragmentation of elongated mitochondria, we decided to apply serial block face scanning electron microscopy (SBF-SEM) data, which is better suited for 3D analysis of mitochondrial length. We found that the length of mitochondria increased at 8 hpi (2.8 ± 0.4 µm) and at 12 hpi (2.9 ± 0.2 µm) in comparison to noninfected cells (2.0 ± 0.1 µm) (S3A and S3B Fig). Altogether, this suggests that the progression of infection triggers mitochondrial elongation accompanied by an increase in the volume *(Fig 2E)*. The cryo-SXT imaging of adherent cells on grids confirmed the infection time-dependent elongation of mitochondria (Fig 2F), and 3D reconstructions of the data revealed increased roughness of the mitochondrial outer surface at 8 and 12 hpi (Fig 2G and S2 Movie). By 12 hpi, the mitochondria had moved closer to the nuclear envelope (Fig 2H) and they became thinner (Fig 2I). Finally, the analysis of the ratio between the mitochondrial surface area and volume confirmed an increase in the surface roughness at late infection (Fig 2J). Taken together, the progression of infection resulted in the increased emergence of elongated and thin mitochondria with a rougher surface that accumulated closer to the nuclear envelope.

## ER-mitochondria interplay is increased in infection

The ER-mitochondria contact sites are formed by protein-mediated tethering of the ER and the outer mitochondrial membrane [50]. Multiple functions that occur at those contact sites include $Ca^{2+}$ signaling, lipid biosynthesis, and mitochondrial division [51]. To observe in high resolution the infection-induced changes in the 3D ultrastructure and specifically in the regions of close contact between the ER and mitochondrial membranes, we used serial block face scanning electron microscopy (SBF-SEM) (Fig 3A). The 3D reconstruction and analyses of the segmented ER and mitochondria from the SBF-SEM data revealed both smaller and more extensive contact sites (Fig 3B and S3 Movie). Infection-induced changes led to a clear decrease in the distance between the ER and mitochondrial membranes at 8 hpi in comparison

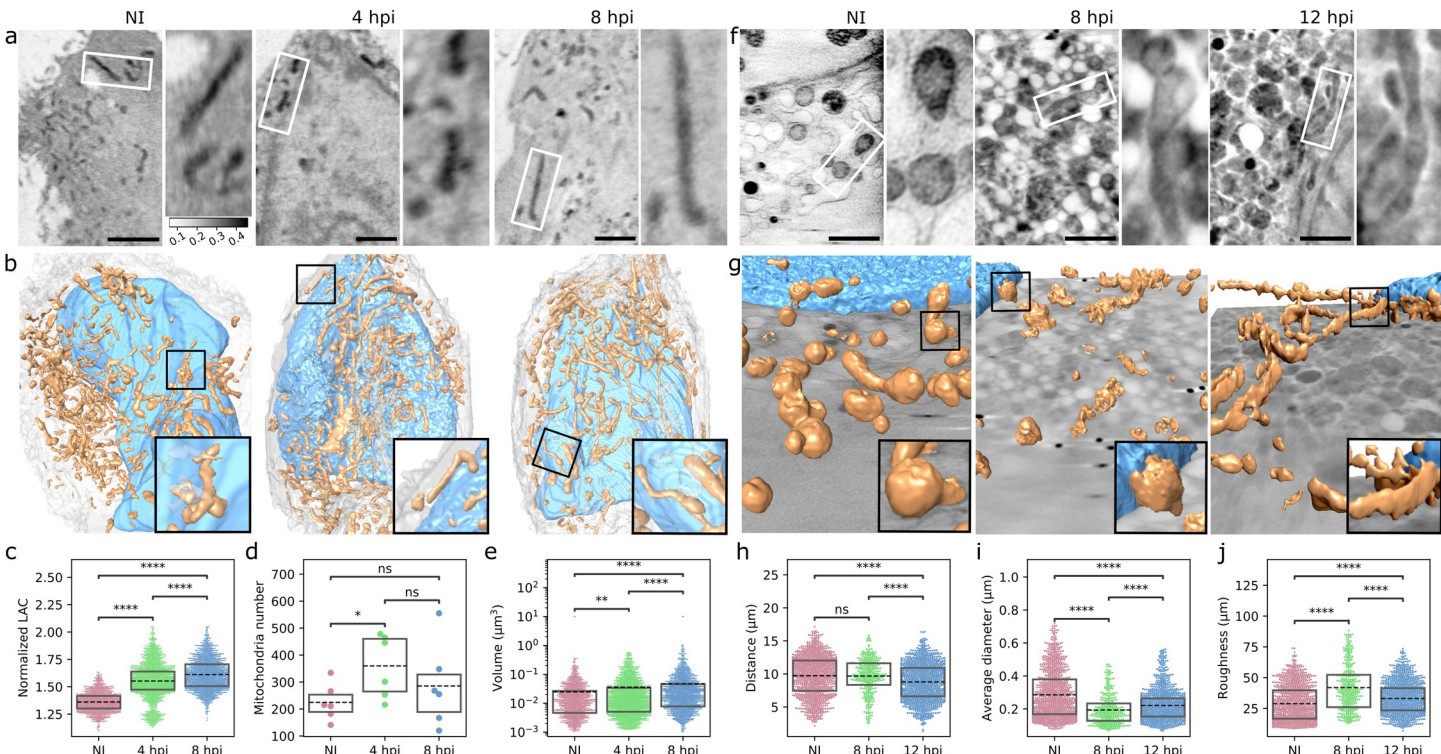

**Fig 2. The progression of infection leads to thinner and rougher mitochondria. (a)** Cryo-soft X-ray tomography (SXT) thin slices (ortho slices) of suspended MEF cells in glass capillaries and **(b)** 3D reconstructions of segmented mitochondria (brown) in the cytoplasm around the nucleus (blue) in noninfected (NI) and infected MEF cells at 4 and 8 hpi. White and black squares show the magnified mitochondria. Scale bars, 2 μm. See also S1 Movie. Box plots of **(c)** linear absorption coefficient (LAC) values (grayscale bar 0.1–0.5 1/microns), **(d)** number, and **(e)** volume of mitochondria in noninfected and infected cells ($n_{cells}$ = 6, $n_{mito}$ = 1349, 2158, and 1713 for NI, 4 and 8 hpi, respectively). **(f)** SXT images of adherent MEF cells grown on EM grids and **(g)** 3D presentations of segmented mitochondria (brown) and nuclear envelope (blue) in noninfected and infected MEF cells at 8 and 12 hpi. The magnified mitochondria are shown in white and black squares. Scale bars, 2 μm. See also S2 Movie. **(h)** The distance of mitochondria from the nucleus, **(i)** the diameter along the short axis, and **(j)** surface-to-volume ratio analysis of mitochondrial roughness in noninfected and infected cells ($n_{cells}$ = 12, 4, and 13, $n_{mito}$ = 1089, 364, and 1165 for NI, 8, and 12 hpi, respectively). The box plots show the mean (dashed line) and the interquartile range. Statistical significance was determined using the Brunner-Munzel test. The significance values shown are denoted as **** (p<0.0001), * (p<0.05), or ns (not significant).

to noninfected control cells. At 12 hpi the distance increased in comparison to 8 hpi, showing that ER and mitochondria move further away from each other in the late stages of infection (Fig 3C). This was supported by an increased number of contact sites and contact region area at 8 hpi and a slight decrease at 12 hpi (Fig 3D and 3E).To study the ER-mitochondria linkage further, we analyzed the distribution of tethering protein located on the ER, namely vesicle-associated membrane protein B (VAPB), by using the ten-fold expansion microscopy [52]. In infected cells at 8 hpi, the VAPB intensity was decreased in comparison to noninfected cells and it was distributed heterogeneously and formed clusters along the mitochondria, in contrast to the more uniform distribution seen in noninfected cells (Fig 4A). VAPB also moved further away from the mitochondria in infected cells when compared to noninfected cells (Fig 4B). Having shown a change in VAPB distribution, we next analyzed ER-mitochondria contact sites by closeness of ER-mitochondria tethering proteins VAPB and regulator of microtubule dynamics 3 (RMDN3, also known as PTPIP51) using proximity ligation assay (PLA). PLA is an immunodetection technique that generates a fluorescent signal only when two antibodies against the antigens of interest are closer than 40 nm from each other [53]. PLA between VAPB and RMDN3 antibodies revealed an increased number of punctate PLA signals as the infection proceeded (Fig 4C and 4D). Notably, the progression of infection also resulted in an increase in the volume of the PLA foci accompanied by an increased size variation (Fig 4E)

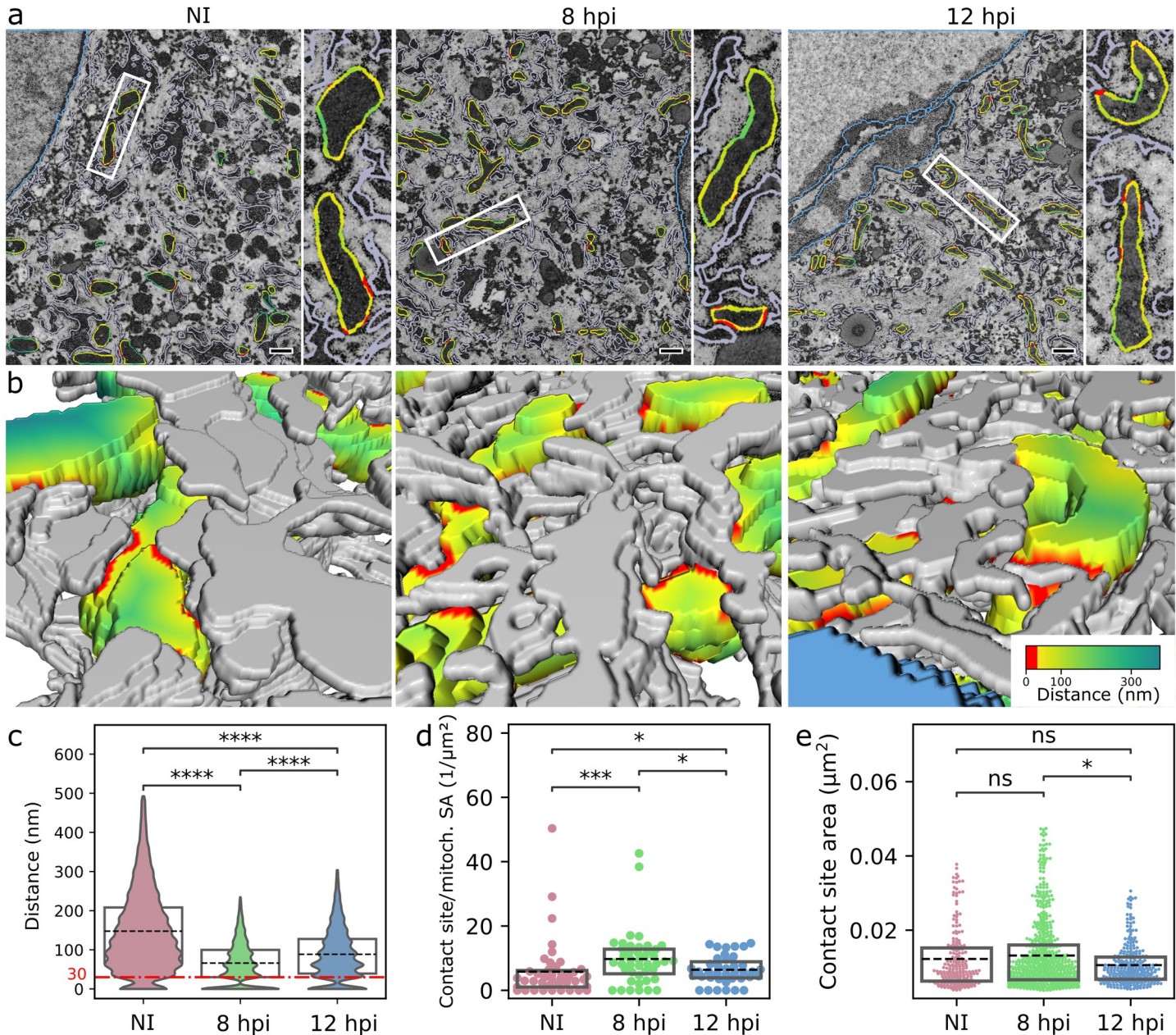

**Fig 3. ER-mitochondria contacts increase in infection. (a)** Representative sections of serial block face scanning electron microscopy (SBF-SEM) of noninfected and infected MEF cells at 8 and 12 hpi. White squares show the magnified mitochondria. The pseudocolor lines around the mitochondria show the closeness of ER and mitochondria, and regions of contact are defined as points where opposed membranes are within 30 nm of each other (red). Scale bars, 0.5 µm. **(b)** Higher-magnification 3D SBF-SEM reconstructions show the regions of contact (distance less than 30 nm, red) between mitochondria (pseudocolor) and ER (grey). The pseudocolor bar indicates the distance between the ER and mitochondria. See also S3 Movie. Violin and box plots show **(c)** the ER-mitochondria distance, **(d)** the number (Nb) of contact sites/mitochondrial surface area, and **(e)** the area of the contact sites ($n_{mito}$ = 43, 43, and 39 for NI, 8, 12 hpi, respectively). The box plots show the mean (dashed line) and the interquartile range. Statistical significance was determined using the Brunner-Munzel test. The significance values are denoted as **** ($p<0.0001$), *** ($p<0.001$), * ($p<0.05$), or ns (not significant).

while being extremely specific (S4 Fig). The presence of enlarged foci suggests that part of the contact sites form discrete clusters on the mitochondrial membrane as the infection proceeds. Overall, these analyses demonstrated that the progression of infection results in increased VAPB-RMDN3 tethering and clustering of ER-mitochondria contact sites.

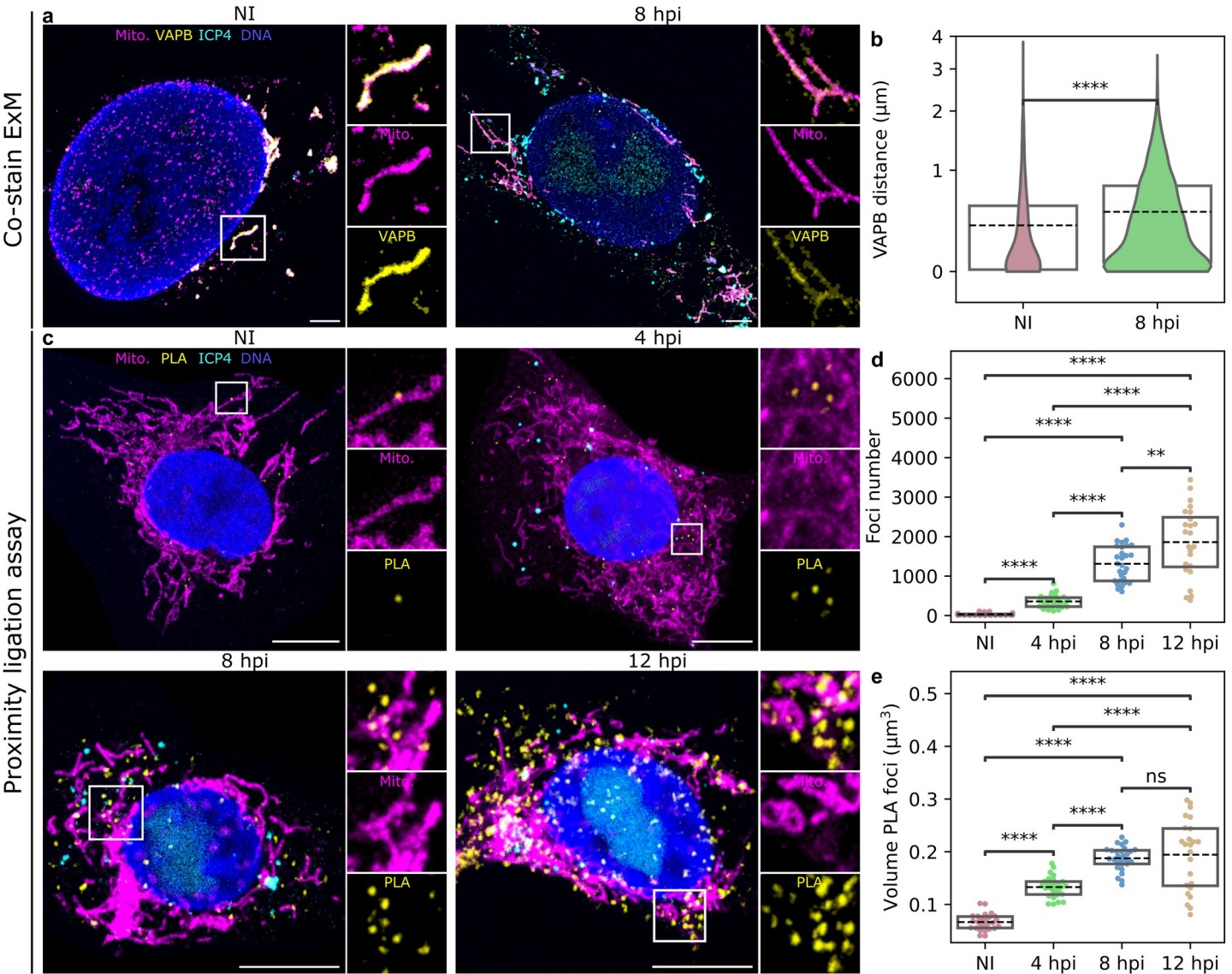

**Fig 4. The number of ER-mitochondria contact sites increase and they are clustered in infection. (a)** Visualization of the mitochondrial structure by the ten-fold robust expansion microscopy (TREx) in noninfected and infected Vero cells at 8 hpi (n = 6). The distributions of an ER protein tyrosine vesicle-associated membrane protein B (VAPB, yellow) located in the ER-mitochondria contact sites and mitochondria labeled with MitoTracker (magenta) are shown. The localization of the nuclear viral replication compartment and the cytoplasmic viral ICP4 protein are presented by HSV-1 EYFP-ICP4 (cyan) and the nucleus by the DAPI stain (blue). Scale bars, 1 μm. (**b**) Violin plots show the distance between the mitochondria and VAPB. (**c**) Proximity ligation analysis (PLA) of contact sites in noninfected and infected Vero cells at 4, 8, and 12 hpi. The PLA signal between VAPB and regulator of microtubule dynamics (RMDN3) is visualized by fluorescent spots (yellow). Mitochondria are labeled with MitoTracker (red), the nucleus with DAPI, and viral replication compartments are visualized by EYFP-ICP4. Scale bars, 10 μm. (**d**) Box plots showing the number of PLA foci and (**e**) volume of the foci per cell (n_cells = 27, 27, 28, and 24 for NI, 4, 8, and 12 hpi, respectively). The box plots show the mean (dashed line) and the interquartile range. Statistical significance was determined using the Student´s t-test. The significance values are denoted as **** (p<0.0001), ** (p<0.01), * (p<0.05) or ns (not significant).

## Mitochondrial cristae thicken and shorten as infection proceeds

Our GRO-seq findings, demonstrating the upregulation of positive regulators and downregulation of negative regulators of cristae formation (Fig 1C), suggested that the progression of infection leads to changes in cristae. We used focused ion-beam scanning electron microscopy (FIB-SEM) to characterize cristae 3D phenotypes as HSV-1 infection progresses in MEF cells (Fig 5A). The mitochondria showed hallmarks of cristae remodeling at 8 hpi, including

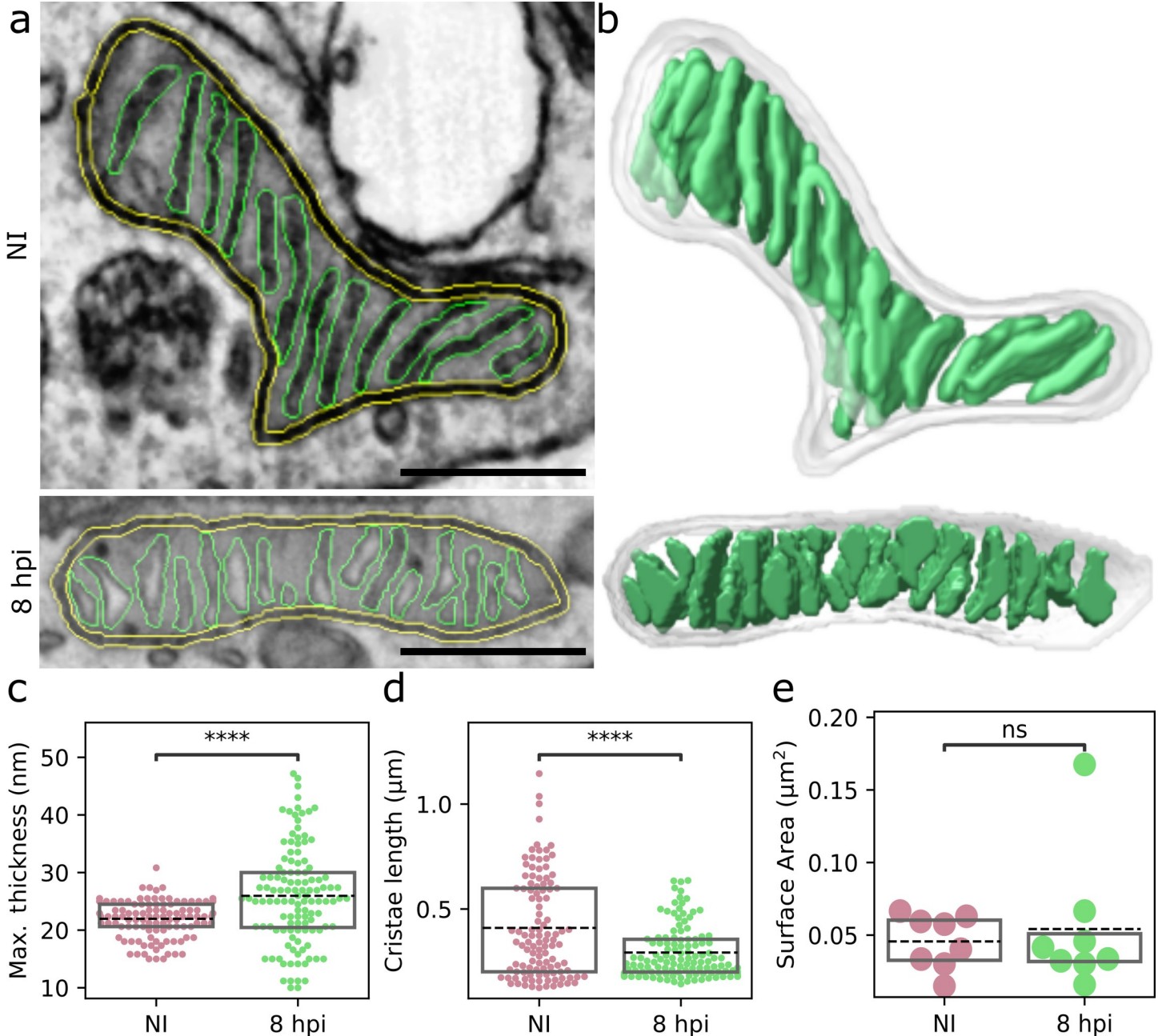

**Fig 5. The cristae become thicker and shorter along the progression of the infection. (a)** Representative focused ion-beam scanning electron microscopy (FIB-SEM) images of noninfected and HSV-1-infected MEF cells at 8 hpi. The mitochondrial outer membrane (yellow) and cristae (green) are shown. Scale bars, 0.5 μm. **(b)** The 3D structure of cristae (green) reconstructed from FIB-SEM stacks. See also S4 Movie. The 3D quantitative analysis of **(c)** the maximal thickness and **(d)** the length of cristae calculated using a watershed algorithm to individualize the cristae lamella in noninfected and infected cells ($n_{mito}$ = 8, $n_{crist}$ = 112 and 121 for NI and 8 hpi, respectively). **(e)** The surface area of segmented cristae in each mitochondria. The box plots show the mean (dashed line) and the interquartile range. Statistical significance was determined using the Student´s t-test. The significance values are denoted as **** ($p<0.0001$) or ns (not significant).

thickening and shortening of cristae (Fig 5B and S4 Movie). Analyses of segmented and reconstructed image data showed that the average thickness of the lamella-shaped cristae increased at 8 hpi when compared to control cells and, specifically, some of the cristae were visibly swollen. In general, the thickness of cristae was more variable in infected than in noninfected cells

(Fig 5A, 5B and 5C). In comparison to noninfected cells, the progress of infection significantly shortened cristae (Fig 5D) without changing their total surface area per mitochondrion (Fig 5E).

## Infection increases mitochondrial proton leakage and Ca²⁺ content

We next asked whether the remodeling of cristae is associated with the modulation of mitochondrial energy metabolism. Oxygen consumption rate (OCR) of the host MEF cells was analyzed using the Seahorse XF24 analyzer (Fig 6A). During the infection, basal respiration and ATP production decreased at 4 hpi and restored to the level of noninfected cells at 8 hpi (Fig 6B and 6C). In contrast, the mitochondrial proton leakage into the matrix through the inner mitochondrial membrane significantly increased only at 8 hpi (Fig 6D). The coupling efficiency, which compares how oxygen is distributed between ATP synthesis and proton leak, demonstrated that ATP synthesis efficiency was best in noninfected cells and decreased towards 8 hpi (Fig 6E). The single-cell fluorescence microscopy measurements revealed that the mitochondrial calcium uptake (Fig 6F) increased at 4 hpi and even more at 8 hpi (Fig 6G), which is consistent with the increase in the number of ER-mitochondria contact sites (Fig 3) mediating Ca²⁺ import to mitochondria from the ER [54]. It has been shown that an increase in calcium, together with oxidative stress, leads to changes in the opening of the mitochondrial permeability transition pore (mPTP) involved in calcium efflux [55,56]. However, our analysis of mPTP status using cellular loading of fluorescent Calcein-AM and Co²⁺ quencher [57] showed that the progression of infection led to an increased cytosolic fluorescence of Calcein-AM. This suggests that instead of opening mPTP the infection appears to favor the closed pore state (S5A and S5B Fig) during the progression of the infection (S5C Fig).

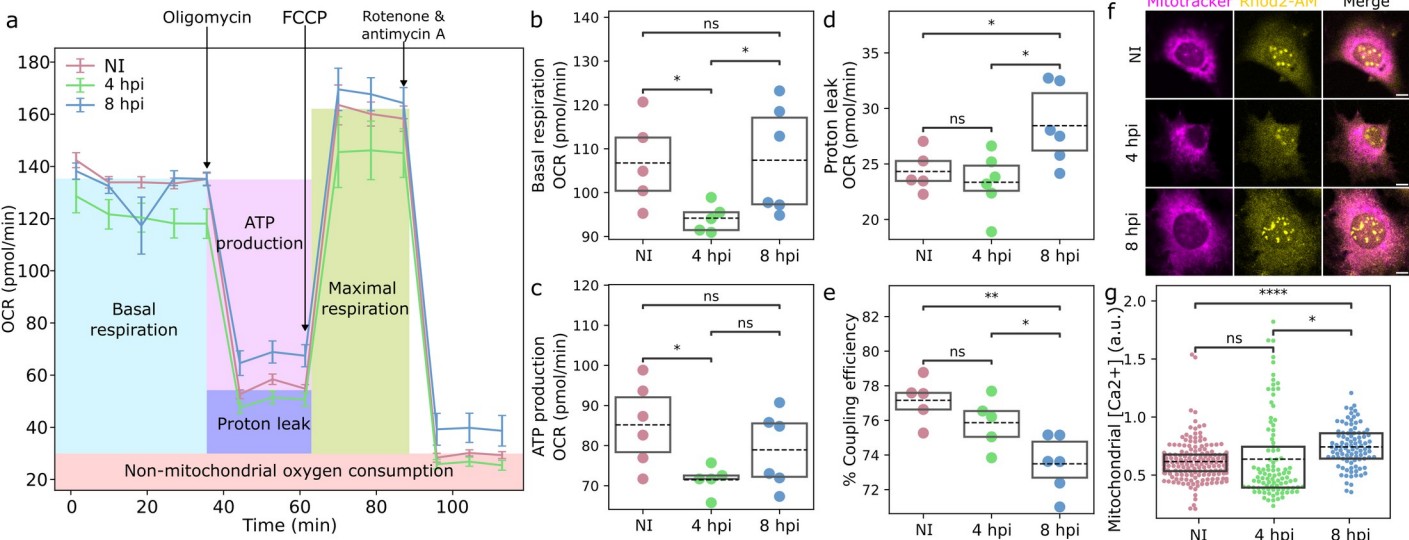

**Fig 6. The infection leads to an increase in the mitochondrial proton leakage and Ca²⁺ content. (a)** Seahorse real-time cell metabolic analysis of oxygen consumption rate (OCR) traces of noninfected and infected MEF cells at 4 and 8 hpi. The arrows mark the addition of oligomycin, FCCP, and rotenone + antimycin A (n = 30 000 cells, 6 replicates). Box plots showing (**b**) basal OCR, (**c**) ATP production, (**c**) proton leak, and (**e**) coupling efficiency (ATP/O). (**f**) The live cell analysis of the mitochondrial calcium labelled by calcium ion indicator Rhod2-AM (green) selectively accumulating within mitochondria, and mitotracker (red). The infected cells selected for imaging were identified by the expression of HSV-1 ICP4 (not shown). Scale bars, 10 μm. (**g**) The quantitative analysis of mitochondrial [Ca²⁺] in noninfected and infected cells at 4 and 8 hpi (n = 207, 97, and 96 for NI, 4, and 8 hpi, respectively). The box plots show the mean (dashed line) and the interquartile range. Statistical significance was determined using the Student's t-test. The significance values shown are denoted as **** (p<0.0001), ** (p<0.01), *(p<0.05), or ns (not significant).

Altogether, ATP production, proton leakage, and mitochondrial [Ca$^{2+}$] were more pronounced at 8 compared to 4 hpi, suggesting a progressive change due to the progression of virus infection.

## Discussion

In infected cells, there is a dynamic balance between viral reprogramming of cellular machinery to advance viral replication and counteracting the infection by cellular defenses. Mitochondrial metabolism and antiviral functions have a central role in this process. Our results demonstrate that the progression of HSV-1 infection results in extensive remodeling of mitochondrial gene transcription, organization, interactions, and energy metabolism.

During the early HSV-1 infection, energy is required for the stepwise progression of viral gene expression, genome replication, and assembly of progeny virions [21,58,59]. We show that the progression of HSV-1 infection from early to later phases results in a reduction in the transcription of genes encoding the proteins of the mitochondrial respiratory chain, specifically the proteins of complex I. Consistent with earlier studies, we show that the mitochondria are reassembled and mitochondrial ATP machinery is reactivated later and remains active until 12 hpi [27,38]. The ATP production declines later, after 18 hpi, when mitochondria are fragmented and mitochondrial DNA is released into the cytosol [27,38,60]. The low cellular level of ATP is one of the factors that lead to apoptosis [61]. The observation of active ATP production at 12 hpi supports the model that apoptosis is activated only at later stages of HSV-1 infection. However, we also show that the cellular proapoptotic processes are supported already at 4 and 8 hpi not only by upregulation of the genes encoding apoptosis-inducing proteins [34,41,42] but also importantly by downregulation genes encoding apoptosis-blocking proteins [36]. Therefore, the preserved ATP production together with viral antiapoptotic factors retain control over the cellular proapoptotic tools during viral assembly and egress, while this dynamic balance likely shifts toward activation of apoptosis as the infection proceeds further.

Our data reveal that the transcription of mitochondrial genes associated with the mitochondrial organization is extensively regulated during the progression of HSV-1 infection. Specifically, the downregulation of the genes encoding proteins involved in mitochondrial fission could lead to mitochondrial elongation (Fig 7A). This is supported by a recent soft X-ray microscopy study also showing that HSV-1 infection leads to mitochondrial elongation [28]. Elongation has previously been observed in the dengue virus infection, in which mitochondrial fusion is induced by the inactivation of mitochondrial fission protein, dynamin-related protein 1 (DRP1) [62]. The balance between mitochondrial fusion and fission machinery is changed during the progression of apoptosis leading to mitochondrial fragmentation and blebbing, activation of caspases, and cytochrome c release from mitochondria [63]. The presence of elongated mitochondria in HSV-1-infected cells at 8–12 hpi supports our other findings that the early events of apoptosis including mitochondrial fission have not yet been activated. Notably, our observation of HSV-1 infection-induced relocalization of mitochondria closer to the nuclear envelope in fibroblast (Fig 7A) is supported by studies in neurons and keratinocytes [27,64]. It has been shown earlier that the cytoplasmic distribution of mitochondria in noninfected cells is regulated by microtubule motor protein contacts with mitochondrial outer membrane protein, mitochondrial Rho GTPase 1 (MIRO1). Deleting MIRO1 causes the repositioning of mitochondria close to the nucleus and results in elevated local concentrations of ATP, H$_2$O$_2$, and Ca$^{2+}$ in the cytoplasm close to the nucleus [65]. MIRO1 is a calcium sensor protein and elevated cytoplasmic [Ca$^{2+}$] results in a decrease in mitochondrial dynamics or motion standstill in neurons [66,67]. The intracellular [Ca$^{2+}$] increases in HSV-1 infection of neural cells and it leads to a significant reduction of MIRO1-mediated mitochondrial mobility

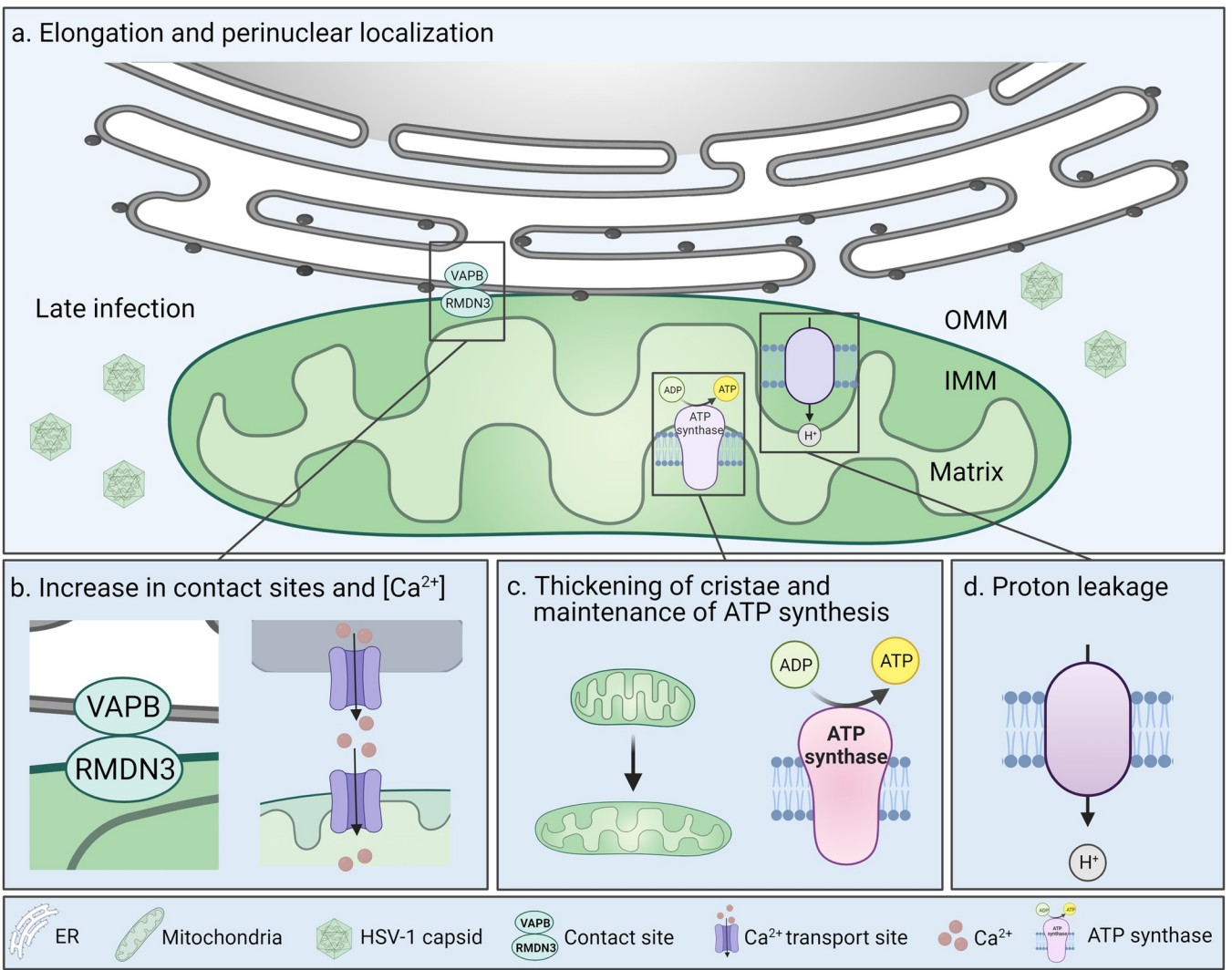

**Fig 7. Mitochondrial structure and function are altered in response to progression of herpesvirus infection.** (**a**) Progression of HSV-1 infection from 4 to 8–12 hpi triggers mitochondrial elongation and repositioning to the perinuclear region. (**b**) The number of the ER-mitochondria membrane contact sites tethered by VAPB and RMDN3 is increased at late infection and mitochondrial $Ca^{2+}$ content is elevated. (**c**) The progression of infection leads to the thickening of mitochondrial cristae and recovery of ATP production to the level of noninfected cells. (**d**) At the same time, the infection stimulates proton leakage across the mitochondrial inner membrane. VAPB, vesicle-associated membrane protein B; RMDN3, regulator of microtubule dynamics 3; OMM, outer mitochondrial membrane; IMM, inner mitochondrial membrane.

at late stages of infection [68]. The HSV-1 infection-induced elevated level of $[Ca^{2+}]$ could also explain the perinuclear distribution of mitochondria in MEF cells. Moreover, the multifunctional HSV-1 ICP34.5 protein which regulates mitochondrial dynamics could have a potential role in the positioning of mitochondria. ICP34.5 interacts with mitochondrial phosphatase PGAM5 which is responsible for the transport and perinuclear localization of mitochondria under infection-induced stress conditions in neural cells [69].

Multiple mitochondrial functions are regulated through the ER-mitochondria membrane contact sites in the outer mitochondrial membrane [70]. Besides being essential for mitochondrial signaling, buffering cytosolic $Ca^{2+}$ level by uptake, division, and metabolism, the contact sites are known to be involved in the infection progress-related balancing of cellular pro- and antiviral responses. Recently Cook et al. elegantly demonstrated that in HCMV, HSV-1,

Influenza A, and beta-coronavirus HCoV-OC43 infection the timely recruitment of ER-mito-chondria linkers (VAPB and RMDN3) is related to the proceeding of the viral replication [71]. The proteomics analysis also showed that the late HSV-1 infection results in an increased number of other ER-mitochondria contact site proteins, ER ribosome-binding protein 1 (RRBP1), and mitochondrial synaptojanin 2 binding protein (SYNJ2BP) [71,72]. Consistently, our volume EM data, expansion microscopy, and PLA interaction analysis revealed an exten-sive growth in the number, volume, and density of contact sites at late infection simultaneously with an increase in mitochondrial membrane roughness and mitochondrial $Ca^{2+}$ content (Fig 7B). The increased availability and clustering of contact sites most likely result in enhanced $Ca^{2+}$ flux from the ER to mitochondria [11,36] thereby explaining the reactivation of respira-tion at late infection and possibly also the perinuclear localization of mitochondria. These events are supported by our results showing that the expression of the mitochondrial pore-forming subunit of calcium voltage-dependent calcium channel subunit (CACNA1B) is upre-gulated in infection.

In HCMV infection the increase in ER-mitochondria contact sites and ER-to-mitochondria transfer of $Ca^{2+}$ are accompanied by cristae remodeling [71,73]. Our studies identified HSV-1 infection-induced upregulation of genes responsible for cristae organization and ATP genera-tion, and downregulation of genes with a negative impact on cristae formation. Our FIB-SEM data also demonstrated that the infection-induced elongated mitochondria contained shorter and thicker lamellar cristae (Fig 7C). The shortening and widening of the cristae give higher overall curvature to them. Notably, it has been shown that the ATP synthases are preferentially located in the more curved areas, and the energetic efficiency of e.g. heart cells is supported by increased curvature of their cristae [74]. This suggests that the structural alteration of the cris-tae shape may support the energy generation in infected cells. Consistently, our analysis of the energetic metabolism verified that ATP synthesis was maintained at 8 hpi (Fig 7C). Previously published studies showed that the interaction of HSV-1 UL16 with the cellular mPTP compo-nent, adenine nucleotide transporter 2 (ANT2), increases ATP generation [75], and the HCMV-induced remodeling of cristae stimulates cellular respiration [76]. Our results indicate that the infection counteracts the calcium influx-induced opening of mPTP [55,56]. This most likely decreases the permeability of the mitochondrial membrane by an unknown mechanism and, thereby, induction of cell death pathways at this stage of infection [77]. We also showed that the HSV-1 infection is accompanied by increased proton leakage into the mitochondrial matrix independent of ATP synthesis (Fig 7D). The activation of ATP production and reduced membrane potential seem to have opposite roles in viral replication. However, in dengue 2 virus infection the proton leak was accompanied by an increased ATP generation [78], and the leakage may also benefit infection as it has been shown to reduce mitochondrial antiviral sig-naling and response [79].

Our multimodal integration of advanced imaging, transcriptomics, and metabolomics draws a comprehensive picture of time-dependent changes in mitochondria as HSV-1 infec-tion proceeds from early to late infection. Our results show how the progression of infection shifts the balance from healthy to diseased cells and leads to profound perturbations in mito-chondrial homeostasis.

## Materials and methods

### Cells and viruses

Mouse embryonic fibroblast cells (*MEF*, ATCC CRL-2991), African green monkey kidney cells (Vero, ATCC CCL-81), and human cervical cancer cells (Hela, ATCC CCL-2). were grown in Dulbecco's modified Eagle medium (DMEM) supplemented with 10% fetal bovine serum, L-

glutamine, and penicillin-streptomycin (Gibco-Invitrogen, Carlsbad, CA) at 37°C in the presence of 5% $CO_2$. The cells were infected 4, 8, or 12 hours before imaging and measurements with wild-type 17+ or EYFP-ICP4 (vEYFP-ICP4) strain [80] using a multiplicity of infection of 5.

## Global run-on sequencing (GRO-seq)

Noninfected and infected Vero and HeLa cells (at 4 and 8 hpi) were suspended in a swelling buffer (10 mM Tris-HCl, 2 mM $MgCl_2$, 3 mM $CaCl_2$, 1 U/ml SUPERase-In and 1 U/ml RNA-sin plus (RNAse inhibitor, Thermo Fisher Scientific, Carlsbad, CA, USA; RNAse inhibitor, Promega, Madison, Wisconsin, USA) and allowed to swell for 5 minutes. Following this, they were centrifuged at $400 \times g$ for 10 minutes, then resuspended in 500 μl of the same buffer, now supplemented with 10% glycerol and RNAse inhibitors. An additional 500 μl of the glycerol-supplemented buffer with 1% Igepal was gradually added to the cell suspension during gentle vortexing.

The nuclei underwent two washes using the swelling buffer enhanced with 0.5% Igepal and 10% glycerol. This was followed by a single wash with a freezing buffer comprising 50 mM Tris-HCl (pH 8.3), 40% glycerol, 5 mM $MgCl_2$, and 0.1 mM EDTA. After counting, the nuclei were centrifuged at $900 \times g$ for 6 minutes, resuspended to yield a concentration of 5 million nuclei/100 μl in freezing buffer, snap-frozen, and stored at -80°C until further use.

The nuclear run-on reaction buffer (NRO-RB) containing 496 mM KCl, 16.5 mM Tris-HCl, 8.25 mM $MgCl_2$, and 1.65% Sarkosyl (Sigma-Aldrich, Steinheim, Germany) was pre-warmed to 30°C. To each milliliter of NRO-RB, 1.5 mM DTT, 750 μM each of ATP and GTP, 4.5 μM CTP, 750 μM Br-UTP (Santa Cruz Biotechnology, Dallas, Texas, USA), and 33 μl of SUPERase Inhibitor (Thermo Fisher Scientific) were added. This supplemented NRO-RB (50 μl) was mixed with 100 μl of the nuclear samples, and the mixture was incubated at 30°C for 5 minutes.

The preparation of GRO-Seq libraries followed the protocol from Kaikkonen et al. [81]. In summary, the run-on products underwent DNAse I treatment (TURBO DNA-free Kit, Thermo Fisher Scientific), base hydrolysis (RNA fragmentation reagent, Thermo Fisher Scientific), end-repair, and immuno-purification using Br-UTP beads (Santa Cruz Biotechnology). Thereafter, a poly-A tailing reaction was performed (PolyA polymerase, New England Biolabs), followed by steps of circularization and re-linearization. The cDNA templates were PCR amplified (Illumina barcoding) for 11–14 cycles and size-selected to range between 180–300 bp. The final libraries were quantified using a Qubit fluorometer (Thermo Fisher Scientific) and readied for 50 bp single-end sequencing on an Illumina Hi-Seq2000 platform (GeneCore, EMBL Heidelberg, Germany). Post-sequencing, GRO-Seq reads were processed using HOMER software [81] to eliminate A-stretches arising from the library preparation and to discard sequences shorter than 25 bp. The quality of the raw sequencing reads was first assessed using the FastQC tool (http://www.bioinformatics.babraham.ac.uk/projects/fastqc/). Bases failing to meet quality standards, specifically those reads where less than 97% of bases had a minimum phred quality score of 10, were trimmed utilizing the FastX toolkit (http://hannonlab.cshl.edu/fastx_toolkit/). For samples sequenced across multiple lanes, data were combined post-quality control. GRO-seq reads were aligned to ChlSab1.1 genome using the bowtie-0.12.7 software [82] while allowing up to two mismatches and up to three locations per read and the best alignment was reported. Differentially expressed genes were detected using ChlSab1.1.95.gtf coordinates, analyzed using RNA.pl workflow of HOMER 4.3 and edgeR v3.2.2. Genes were considered differentially expressed using the following cutoffs: log2 fold change 1, RPKM at least 0.5 in at least one sample, and adjusted p-value of 0.05. GRO-seq reads were mapped to the human protein atlas (https://www.proteinatlas.org/) to select

mitochondrial proteins. The interaction of these proteins was further mapped using the String database (https://string-db.org/) by selecting experimentally shown interaction using a threshold of 500. The unregulated genes were identified in the same String database with the condition that they interact with at least two regulated genes. The resulting graph was generated in Python using the module network and manually adjusted.

## Cryo-soft X-ray tomography (SXT) of suspended cells in capillaries

MEF cells were seeded into culture flasks (Corning, Corning, NY) and infected with HSV-1 vEYFP-ICP4 at an MOI of 5 for 4 and 8 h at 37˚C. Noninfected and infected cells were detached with trypsin-EDTA, pelleted by centrifugation (300xg), and washed with PBS. Cells were then fixed with glutaraldehyde (0.5%) and paraformaldehyde for 10 minutes at RT and 35 minutes at 4˚C (4%, cat. No. 15713-S; Electron Microscopy Sciences, Hatfield, PA), washed once with PBS, pelleted by centrifugation (300x*g*), re-suspended in the PBS buffer, filtered, loaded into thin-walled cylindrical borosilicate glass capillaries (Hilgenberg GmbH, Hilgenberg, GER, Cat.No. 4023088), and then vitrified by quickly plunging them into liquid-nitrogen-cooled (~90 K) liquid propane. SXT tomographic data were collected by full-rotation imaging with a soft X-ray microscope (XM-2) in the National Center for X-ray Tomography at the Advanced Light Source (http://www.als.lbl.gov) of Lawrence Berkeley National Laboratory (Berkeley, CA). To avoid radiation damage cells were kept in a stream of liquid-nitrogen-cooled helium gas during data collection. For each data set, 180˚ rotation tomographs (1 projection image per 2˚ angle) were collected [83,84]. Projection images were aligned by tracking fiducial markers on adjacent images and reconstructed using automatic reconstruction software [85]. The segmentation of mitochondria and their analyses of soft X-ray tomography datasets were performed manually by using commercial software Amira-Avizo (Thermo Fisher Scientific) based on the morphology or LACs.

## Cryo-SXT of adherent cells on grids

MEF cells were grown and infected with HSV-1 vEYFP-ICP4 at an MOI of 5 for 8 and 12 h at 37˚C on Quatifoil-coated Au-TEM finder grids (R2/2, G200F1, Gilder, Jena, GER). The grids were vitrified by plunging into liquid ethane using Leica EM GP2 (Leica microsystems GmbH, Germany) and kept at ~ -196˚C until imaging. The infected EYFP-ICP4-expressing cells were identified by an online fluorescent microscope. The tomographic tilt-series of infected cells were collected by transmission soft X-ray microscope (MISTRAL beamline, ALBA synchrotron light source, Barcelona, Spain, https://www.cells.es/en) [86,87] at 520 eV with exposure times of 2–4 s and by compact soft X-ray microscope (SXT100, Sirius XT, Dublin, IRE, https://siriusxt.com/) [88]. The projection images corresponding to the typical maximum tilt angles of ±67 degrees were collected for each data set. To create a 3D tomographic reconstruction of the cell, the projection images were normalized by a flat field image, aligned using the fiducial markers (IMOD software [89]) or by patch tracking followed by reconstructing of the aligned stacks using the Simultaneous Iterative Reconstruction Technique (SIRT) algorithm. The mitochondria were segmented using Biomedisa [90] and analyzed by using X-ray absorption threshold analysis [91].

## Serial block face scanning electron microscopy (SBF-SEM)

MEF cells were grown and infected with HSV-1 17+ strain at an MOI of 5 for 8 and 12 h at 37˚**C** on glass coverslips. After fixation (2% glutaraldehyde, 2% paraformaldehyde, 2 mM CaCl$_2$ in 0.1 M sodium cacodylate buffer, pH 7.4, for 30 minutes), the contrast of cells was increased by the double osmication and "en bloc" staining. First osmication was done with 2%

osmium tetroxide, 1.5% potassium ferrocyanide, and 2 mM $CaCl_2$ in 0.1 M sodium cacodylate buffer (pH 7.4) followed by incubation with 1% thiocarbohydrazide. The second osmication was performed by incubating with 1% osmium tetroxide in distilled water. The "en bloc" staining was performed by incubation with 1% uranyl acetate, followed by staining with lead aspartate. The cells were then dehydrated with graded series of ethanol and acetone, and gradually embedded into a Durcupan resin (Sigma-Aldrich, product number 44610) [92]. Prior polymerization the samples were placed on top of a hand-punched silicone sheet clamped between two pieces of Aclar films (Agar Scientific) and two objective glasses. After detaching the specimen from the glass, few square millimeters in area were cut and mounted on an aluminum specimen pin using silver glue. A pyramid was cut using a razor blade and the entire surface of the specimen was then sputtered with a thin layer of platinum to improve conductivity and reduce charging during the imaging. Ultra-thin sections (slice thickness, 30 nm) were cut by ultra-microtome and the newly revealed block faces were imaged by a Quanta 250 (Thermo Fisher Scientific), equipped with Gatan 3View2 (Gatan, Pleasanton, CA). The datasets were acquired with a voxel size of 8 x 8 x 30 $nm^3$ at the accelerating voltage of 2.5 kV, spot size 3.0 and pressure of 0.15 Torr. The data was aligned and processed using Microscopy Image Browser (MIB) [93]. The mitochondria and ER were segmented manually, and ER-mitochondria contact sites were analyzed from the 3D data using a masked distance map.

### Focused Ion Beam Scanning Electron Microscopy (FIB-SEM)

Mitochondrial cristae were imaged using FIB-SEM on MEF cells processed by the ChromEM method [94]. Briefly, non-infected and infected cells (17+ strain) were fixed at 8 hpi with 2.5% glutaraldehyde (EM-grade) in 5 mM $CaCl_2$, 0.1M sodium cacodylate buffer, pH 7.4, at room temperature for 5 minutes followed by 4°C incubation for 55 minutes. Further steps were carried out at 4°C. The cells were washed five times with cacodylate buffer followed by incubation with blocking buffer (10 mM glycine, 10 mM potassium cyanide in 0.1 M sodium cacodylate buffer). Cellular DNA was stained with 10 μM DRAQ5 in 0.1% saponin and 0.1 M sodium cacodylate buffer for 10 minutes followed by washing three times with blocking buffer. Ice cold 2.5 mM diaminobenzidine (DAB) solution was added to the cells and they were exposed to 620 ± 30 nm light from mercury short-arc lamp of Leica EL6000 using Y5ET filter set and 40 X air objective of Leica DM IL LED microscope for 6 minutes, causing the DAB to polymerize over DRAQ5-stained DNA. After rinsing with sodium cacodylate buffer, cells were incubated with 1% osmium tetroxide, 2 mM $CaCl_2$, and 1.5% potassium ferrocyanide in 0.15 M cacodylate buffer, on ice, for 1 hour. After washing with double-distilled water, cells were dehydrated and flat embedded in epoxy resin (TAAB 812, Aldermaston, UK) [95]. A few cubic millimeters sample from the block was cut and mounted on an aluminum specimen pin using silver glue. After cutting a pyramid with a razor blade, the specimen was coated with a thin layer of platinum. The 3D volume datasets were acquired using a Zeiss Crossbeam 550 (Gemini 2 optics, Carl Zeiss Microscopy GmbH, Jena, Germany) and both backscattered and secondary electron signals were collected using inlens detectors. For a FIB-SEM run a thin platinum carbon pad film was deposited on the region of interest of the block using a focused ion beam and a trench was milled in front of the target cell. The cross-sectional area of the cell of interest was then exposed by ablation of the resin by the beam (milling with a 0.7 nA beam current at the acceleration voltage of 30 kV) and the scanning electron beam (0.31 or 0.50 nA, 1.5 or 1.6 keV) imaged the exposed areas, thus creating an imaged volume with a voxel size of 2.5 x 2.5 x 5 $nm^3$. The datasets were collected and aligned using Zeiss Atlas 5 software with 3D tomography module. The mitochondrial cristae were semiautomatically segmented using the segmentation tools of MIB. The local thickness of the cristae was measured using ImageJ

algorithm (https://imagej.net/imagej-wiki-static/Local_Thickness) implemented in Python so that the maximum thickness value was returned. The surface area was calculated using the marching cubes algorithm. The length of the cristae lamella was measured from the watershed cristae after reducing the image to a 2D plane. The longest distance between 2D points was measured.

## Expansion microscopy

Vero cells were grown and infected (HSV-1 vEYFP-ICP4, MOI 5) on glass coverslips. At 8 hpi the infected and non-infected control cells were fixed with 4% PFA (Electron Microscopy Sciences, SKU 15710, diluted to 4% with sterile PBS) for 20 min at room temperature. Before fixation, the cells were labeled using MitoTracker Deep Red (Invitrogen, catalog number M22426) for 60 minutes at 1 µM concentration. The fixed samples were permeabilized for 15 min at RT using 0.5% Triton X-100 in PBS supplemented with 0.5% bovine serum albumin (BSA) and immunolabelled using VAPB antibody (Proteintech, catalog number 66191-1-Ig) with 1:100 dilution and GFP antibody (Abcam, Ab290) with 1:200 dilution that detects the EYFP in HSV-1 ICP4. Primary antibodies were incubated for 2 h at RT after which the samples were washed in PBS three times for 15 minutes. Secondary antibody labeling was done for 1 h at RT using anti-mouse Alexa 488 (1:100) and anti-rabbit Alexa 546 (1:250). Both primary and secondary antibodies were diluted in 3% BSA-PBS.

After immunolabelling a ten-fold robust expansion was performed as described earlier [52]. In short, the samples were anchored using 0.1 mg/ml Acrolyl-X (Invitrogen, Catalog number A20770) overnight at RT. Samples were gelated with a gel containing 14.2% acrylamide (Sigma-Aldrich, product number A9099), 10.1% sodium acrylate (AmBeed, catalog number A107105), 0,0006% bis-acrylamide (Sigma-Aldrich, product number M1533), 0.15% ammonium persulfate (Sigma-Aldrich, A3678), and 0.15% TEMED (Sigma-Aldrich, product number T22500) in PBS for 1 hour at 37°C. A lower crosslinker concentration in comparison to the original publication was used for a higher expansion factor. Gelated samples were denatured in 0.2 M NaCl, 50 mM Tris (pH 6.8), and 5.76% SDS for 1.5 hours at 78°C. Before expansion, the DNA was labeled using DAPI diluted 1:6000 in PBS for 45 minutes. Gels were expanded in ddH$_2$O until no further expansion was observed. The expansion factor was calculated by comparing the size of the gel after the gelation and post-expansion. The achieved expansion was approximately 12-fold which was taken into account in analyses.

The expanded samples were imaged with Leica SP8 using 63X, 1.2 NA water immersion objective. DAPI was excited with a 405 nm laser and detected 412–497 nm with a PMT detector. Alexa 488 was excited with a 499 nm laser and detected 506–551 nm using a Leica HyD SMD detector with a lifetime gate of 0.1–6 ns in a photon counting mode. Alexa 546 was excited with a 553 nm laser and detected 560–585 nm using a Leica HyD SMD detector with a lifetime gate of 0.1–6 ns in a photon counting mode. MitoTracker deep red was excited with a 647 nm laser and detected 654–782 nm using a Leica HyD SMD detector without a lifetime gate in a photon counting mode. A voxel size of 160 x 160 x 366 nm (XYZ) was used. To analyze the VAPB signal, the nucleus was segmented after a Gaussian blur using the Otsu thresholding method, then the binary mask was expanded with a (7x7x7) mask, filled the holes in it and eroded with a similar mask to have the full nucleus. This mask was then used to exclude any signal of VAPB and MitoTracker from inside the nucleus. The VAPB and MitoTracker channels were thresholded with any pixel having a value higher than 0 after a Gaussian blur. The geometric centers of each VAPB foci were calculated and the distance from these geometric centers to the closest mitochondria voxel was calculated.

## Proximity ligation assay

Vero cells were grown and infected with HSV-1 vEYFP-ICP4 at an MOI of 5 at 37°C on glass coverslips. The cells were fixed at 4, 8, and 12 hpi with 4% PFA in PBS and permeabilized with 0.1% Triton X-100 in PBS. The assay [53] was performed with Duolink In Situ detection (DUO9210, orange, Mouse/Rabbit kit, Sigma-Aldrich). Antibodies detected by the PLA were VAPB MAb, paired with Rabbit RMDN3 Ab (HPA009975, Atlas antibodies, Bromma, Sweden). Controls included noninfected controls and noninfected technical controls where the assay was performed either without probes or only with one of the antibodies used. The PLA assay was done according to the manufacturer's protocol. The mitochondria were stained with a Mitotracker Deep Red FM (Invitrogen).

PLA signals were detected with Leica TCS SP8X Falcon confocal microscope (Leica Microsystems, Mannheim, Germany) with HC PL APO CS2 glycerol immersion objective (NA 1.3). Infected cells were identified by the presence of EYFP-ICP4, a marker of the viral replication compartment. DAPI was excited with a 405 nm diode laser and emission between 410–519 nm was detected. EYFP, PLA foci, and MitoTracker were excited with 514 nm, 554 nm, and 641 nm wavelengths of the pulsed white light laser (80 mHz) and detected at 519–549 nm, 559–636 nm, 646–779 nm, respectively. The imaged stacks of single cells were of size 1168 by 1168 pixels with a pixel size of 63 nm and a z-sampling distance of 180 nm.

The nuclei were segmented using the Otsu thresholding and used as a mask to exclude any signal inside the nucleus from the analysis. The mitochondria were segmented from MitoTracker images after passing a Gaussian blur by using automatic Otsu segmentation. The Triangle algorithm automatically segmented the PLA signals after applying a Gaussian blur. A watershed algorithm was applied to separate very close PLA foci. The geometric centers and volume of the PLA signals were calculated, and the distance from the geometric centers to the closest mitochondria voxel was calculated.

## Analyses of mitochondrial functions

The metabolic phenotype of infected cells was determined by extracellular Seahorse flux analysis. Oxygen consumption rate (OCR) analysis was used to address the mitochondrial substrate dependency and maximal respiration levels. Briefly, MEFs were plated on Seahorse 24-well plates $2 \times 10^5$ cells per well and incubated overnight in a complete DMEM $CO_2$-free incubator at 37°C. Cells were infected with HSV-1 EYFP-ICP4 strain (MOI = 5) in a fresh medium and incubated for 4 and 8 hours before measurements. Seahorse Mito stress test assay of noninfected and infected cells was performed using the Seahorse XF Cell Mito Stress Test Kit (103015–100, Agilent, Santa Clara, CA). The final concentrations of drugs were: oligomycin, 1 μM; carbonyl cyanide-p-trifluoromethoxyphenylhydrazone (FCCP), 1 μM; rotenone, 1μM + antimycin A, 1 μM. The experiments were performed according to the manufacturer's instructions.

For single-cell measurements of the mitochondrial calcium, MEF cells (100,000 cells) were seeded on 12 well plates containing 12 mm coverslips 2 days before the experiment. Cells were infected with HSV-1 EYFP-ICP4 strain (MOI = 5) in a fresh medium and incubated for 4 and 8 hours before measurements. One hour before observation, cells were incubated in DMEM (without Phenol Red) supplemented with Rhod2-AM (1μM, Thermo Fisher Scientific), MitoTracker Deep Red (0.5 μM, Thermo Fisher Scientific), and Hoechst 33342 (1 ug/ml, Invitrogen). *Rhod-2*-AM is a lipid-soluble probe that has a weak positive charge that will lead to its accumulation into the polarized mitochondrial matrix. After 30 min incubation at 37°C and 5% $CO_2$, cells were washed twice with warm PBS and further incubated for 30 min in DMEM without Phenol Red. At the indicated time point, the cells were imaged using a confocal

microscope (Nikon A1 plus) using a 20x (NA 0.75) objective lens, single plane, and auto-focus system to acquire a large area (4x4 images) with 10% overlap. Hoechst, Rhod-2-AM and Mitotracker were excited with a 405, 561 and 638 nm laser respectively. Hoechst and Mitotracker were acquired simultaneously with an emission center to 450 and 705 nm respectively while Rhod-2 was acquired with a 595/50 filter cube on line scan mode. One image was 1024x1024 pixels with a pixel size of 340nm, for a total recombined image of 3789x3789 pixels. After tiling the images, the nuclei were segmented, infected cells were automatically selected, and the Rhod-2 signal was normalized by the intensity signal of the mitochondria.

To study the opening of the mPTP pores during the infection, we measured the cytoplasmic intensity of Calcein AM staining in noninfected and infected cell. A Mitochondrial permeability transition pore assay kit was used (333-K2061-100, APExBIO, Houston, TX). MEF cells (175,000 cells) were grown on live imaging plates overnight. Before the experiment, cells were infected with HSV-1 EYFP-ICP4 virus with MOI 5. The experiment was performed at 4, 8, and 12 hpi by adding a working solution containing fluorescent Calcein AM and a $CoCl_2$ quencher. The solution was supplemented with MitoTracker Deep Red (0.5 μM, Thermo Fisher Scientific) to visualize the mitochondria. The same areas of the samples were imaged straight after adding the working solution and after 30 min incubation to detect the infected cells. EYFP (Calcein AM and HSV-1 EYFP-ICP4) and Mitotracker deep red were acquired with 514 nm and 641 nm wavelengths of the pulsed white light laser (80 mHz). The signals were detected with HYD detectors at 519–636 nm and 646–779 nm in photon counting mode, respectively. Stacks of 512 x 512 pixels were acquired with a pixel size of 300 nm in the xy and 1 um in z. The images were merged and maximum projected to obtain a single plane image of 2360x2360 pixels. For the image analysis, the images at time 0- and 30-min incubation were used. First, the images were smoothed with a gaussian filter (sigma = 2), the cell cytoplasm detected by using a local threshold on the mitochondria channel, and from that the negative of this image gives the nucleus. After removing too small nuclei (less than 300 pixels), each nucleus was paired with a watershed cytoplasm if their centroids did not differ much. On each cell, the green fluorescence was measured at time 0 min in the nucleus, giving the infection status, as well as in the cytoplasm only giving the ICP4 cytoplasmic contribution. This signal was then subtracted from the Calcein signal in the cytoplasm.

## Supporting information

**S1 Fig. Cellular and mitochondrial proteins regulated in response to viral infection. (a)** A Venn diagram of the GRO-Seq dataset showing gene regulation of mitochondrial protein-associated gene transcription in infected Vero cells. The number of regulated proteins at 4 (green) and 8 hpi (pink), or at both time points (blue) is shown. The size of the circle is proportional to the number of regulated proteins. (**b**) The upregulation and downregulation of mitochondrial genes at 4 and 8 hpi clustered according to their predicted functional protein-protein interactions. GO terms of their functional classes of major biological processes detected in infection are shown. The horizontal line at value 0 represents gene transcription in noninfected cells, and positive and negative values represent the upregulation and downregulation of the gene transcription.
(TIF)

**S2 Fig. Analysis of nascent RNA levels of mitochondrial and mitochondria-associated proteins in infected Hela cells. (a)** A Venn comparison of GRO-Seq dataset showing gene regulation of mitochondrial protein-ssociated gene expression in infected Hela cells at 4 and 8 hpi. The number of regulated proteins at 4 (green) and 8 hpi (pink), or at both time points (blue) is shown. The size of the circle is proportional to the number of regulated proteins. (**b**) The

upregulation and downregulation of mitochondrial genes at 4 and 8 hpi clustered according to their predicted functional biological process during infection. The horizontal line at value 0 represents gene expression in noninfected cells, and positive and negative values represent the upregulation and downregulation of gene expression, respectively.
(TIF)

**S3 Fig. Mitochondria elongate as the infection proceeds. (a)** Serial block face scanning electron microscopy (SBF-SEM) images of noninfected and infected MEF cells at 8 and 12 hpi. Scale bars: 2 μm. (**b**) Quantitative analysis of mitochondrial length ($n_{mito}$ = 202, 64, and 244 for NI, 8, and 12 hpi, respectively). The box plots show the mean (dashed line) and the interquartile range. Statistical significance was determined using the Student's t-test. The significance values are denoted as **** (p<0.0001), *(p<0.05), or ns (not significant).
(TIF)

**S4 Fig. Technical controls in the PLA assay.** Representative images of the PLA technical controls showing the PLA signals in noninfected cells either without the PLA probes (left), without the VAPB antibody (middle), or without the RMDN3 antibody (right). The PLA signals between VAPB and RMDN3 are visualized in yellow. Mitochondria are labeled with Mito-Tracker (magenta), the nucleus with DAPI (blue), and EYFP-ICP4 is used as the viral marker (cyan). The marked areas are magnified on the right side of each image, showing the PLA signal together with the mitotracker, only the mitotracker, and only the PLA signal, respectively. Scale bars, 10 μm.
(TIF)

**S5 Fig. Mitochondrial permeability transition pores prefer closed state in infected cells.** The state of mPTP was assessed by cellular loading of fluorescent Calcein-AM and $Co^{2+}$ quencher. (**a**) Representative cells showing Calcein (cyan) distribution in the cytosol of noninfected and infected MEF cells at 4, 8, and 12 hpi. The plasma membrane localization is shown (cell, grey line). (**b**) The fluorescent intensity of cytoplasmic Calcein and nuclear viral replication compartment marker, ICP4, in noninfected and infected cells at 4, 8, and 12 hpi (n = 150, 134, 65, and 152, respectively). Statistical significance was determined using the Student´s t-test. The significance values are denoted as **** (p<0.0001), *** (p<0.001), or ns (not significant).
(TIF)

**S1 Table. Mitochondrial and mitochondria-associated protein genes are regulated in response to viral infection in Vero cells.** The table presents the results of GRO-seq analysis showing upregulated (yellow-red gradient) or downregulated (yellow-green gradient) mitochondrial and mitochondria-associated protein genes in infected Vero cells at 4 and 8 hpi.
(XLSX)

**S2 Table. Enrichment of biological pathways of mitochondrial and mitochondria-associated protein genes regulated in infection.** GO functional annotation chart of mitochondria-connected regulated protein genes created by the PANTHER classification system for GO Biological process.
(XLSX)

**S3 Table. Mitochondrial and mitochondria-associated protein regulation in infected HeLa cells.** The table presents the results of GRO-seq analysis showing upregulated or downregulated mitochondrial and mitochondria-associated protein genes in infected Hela cells at 4, 8 and 12 hpi.
(XLSX)

**S4 Table. Microscopy techniques.** Microscopy techniques, sample preparation, data analyses, and comparison of techniques used in the studies.
(XLSX)

**S1 Movie. Structural changes in mitochondria of HSV-1 infected cells.** Cryo SXT images and their 3D reconstructions of suspended noninfected and infected MEF cells at 4 and 8 hpi. The localization of the nucleus (blue) and mitochondria (light brown) are shown.
(MOV)

**S2 Movie. HSV-1 infection-induced alterations of mitochondrial morphology.** Cryo SXT images and their 3D reconstructions of adherent noninfected and infected MEF cells at 8 and 12 hpi. The localization of the nucleus (blue), mitochondria (light brown), lipids droplets (yellow), and multivesicular bodies (green) are shown.
(MOV)

**S3 Movie. ER-mitochondria contact sites increase in HSV-1-infected adherent MEF cells.** Serial block face scanning electron microscopy (SBF-SEM) images and reconstructions showing ER-mitochondria contact sites in noninfected and infected MEF cells at 8 and 12 hpi. The pseudocolor scale of mitochondria indicates its distance (less than 30 nm, red) from the ER (grey).
(MOV)

**S4 Movie. The structural changes in mitochondrial cristae as the infection proceeds.** Focused ion-beam scanning electron microscopy (FIB-SEM) images and reconstructions of cristae (green) in noninfected and infected MEF cells at 8 hpi.
(MOV)

**S1 Data. The data for Figs 2–6.**
(XLSX)

## Acknowledgments

We thank the staff of the National Center for X-ray Tomography (NCXT) and Advanced Light Source (Lawrence Berkeley National Laboratory, Berkeley, CA), the MISTRAL (BL09) beamline at ALBA Synchrotron (Barcelona, Spain), Diamond Light Source (Oxford, UK), and SiriusXT Limited (Dublin, Ireland) for providing cryo-SXT experiments. We thank Arja Strandell, Mervi Lindman, and Mervi Laanti for technical assistance with EM samples, and the Electron Microscopy Unit (EMBI) of the Institute of Biotechnology, University of Helsinki (Helsinki, Finland) for EM sample preparation and volume EM imaging. We thank the technical support of Sami Salminen, Jänne Kärnä, and Niklas Kähkönen with Seahorse experiments done in BioMediTech (Faculty of Medicine and Health Technology, Tampere University, Tampere, Finland), David Rogers, Stephen O'Connor (SiriusXT) for imaging of cryo-SXT samples, and Niilo Joutsenlahti for GRO-seq sample preparation. We acknowledge Biocenter Finland for infrastructure support. Fig 7 was created with BioRender.com.

## Author Contributions

**Conceptualization:** Simon Leclerc, Vesa Aho, Maija Vihinen-Ranta.

**Data curation:** Simon Leclerc, Henri Niskanen, Ilya Belevich, Minna U. Kaikkonen.

**Formal analysis:** Simon Leclerc, Axel A. Ekman, Henri Niskanen, Minna U. Kaikkonen, Venera Weinhardt.

**Funding acquisition:** Eva Pereiro, Tony McEnroe, Carolyn A. Larabell.

**Investigation:** Simon Leclerc, Alka Gupta, Visa Ruokolainen, Jian-Hua Chen, Kari Kunnas, Henri Niskanen, Ana J. Perez-Berna, Sergey Kapishnikov, Elina Mäntylä, Salla Mattola.

**Methodology:** Jian-Hua Chen, Helena Vihinen, Paula Turkki.

**Project administration:** Maija Vihinen-Ranta.

**Resources:** Maria Harkiolaki, Eric Dufour, Vesa Hytönen, Eva Pereiro, Tony McEnroe, Kenneth Fahy, Eija Jokitalo, Carolyn A. Larabell.

**Software:** Simon Leclerc, Axel A. Ekman, Ilya Belevich, Venera Weinhardt, Vesa Aho.

**Visualization:** Simon Leclerc, Alka Gupta, Visa Ruokolainen, Jian-Hua Chen, Sergey Kapishnikov, Salla Mattola.

**Writing – original draft:** Maija Vihinen-Ranta.

**Writing – review & editing:** Helena Vihinen, Vesa Hytönen, Vesa Aho, Maija Vihinen-Ranta.

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
