## [Decision Letter · Decision Letter 0]

2 Jan 2024

Dear Adjunct professor Vihinen-Ranta,

Thank you very much for submitting your manuscript "Progression of herpesvirus infection remodels mitochondrial organization and metabolism" for consideration at PLOS Pathogens. As with all papers reviewed by the journal, your manuscript was reviewed by members of the editorial board and by several independent reviewers. In light of the reviews (below this email), we would like to invite the resubmission of a significantly-revised version that takes into account the reviewers' comments.

Dr. Vihinen-Ranta,

I hope this finds you well. Thank you for your recent submission to PLoS Pathogens. Your manuscript was reviewed by three experts in the Herpesvirus field. While all were supportive of the study, some significant major concerns were raised. It is the opinion of the editors that these proposed experiments and alterations suggested by the reviewers are necessary to render the submission suitable for publication. As such, a decision of Major Modify has been agreed upon by the editors. If you are able to address all the significant concerns in a timely fashion, we would reconsider the submission.

Cheers,

Eain A. Murphy.

We cannot make any decision about publication until we have seen the revised manuscript and your response to the reviewers' comments. Your revised manuscript is also likely to be sent to reviewers for further evaluation.

Sincerely,

Eain A Murphy, Ph.D.

Academic Editor

PLOS Pathogens

Patrick Hearing

Section Editor

PLOS Pathogens

Kasturi Haldar

Editor-in-Chief

PLOS Pathogens

orcid.org/0000-0001-5065-158X

Michael Malim

Editor-in-Chief

PLOS Pathogens

orcid.org/0000-0002-7699-2064

Dr. Vihinen-Ranta,

I hope this finds you well. Thank you for your recent submission to PLoS Pathogens. Your manuscript was reviewed by three experts in the Herpesvirus field. While all were supportive of the study, some significant major concerns were raised. It is the opinion of the editors that these proposed experiments and alterations suggested by the reviewers are necessary to render the submission suitable for publication. As such, a decision of Major Modify has been agreed upon by the editors. If you are able to address all the significant concerns in a timely fashion, we would reconsider the submission.

Cheers,

Eain A. Murphy.

Reviewer's Responses to Questions

**Part I - Summary**

Reviewer #1: In the manuscript PP-D-23-02006, "Progression of herpesvirus infection remodels mitochondrial organization and metabolism", Leclerc et al, performed transcriptomic, microscopic, and metabolomic analyses of mitochondrial changes during host cell infection by HSV-1. Using global run-on (GRO) sequencing of uninfected Vero cells in comparison with those infected for 4 or 8h with HSV-1, they find many changes in transcription of mitochondrial and mitochondrial-associated genes. Gene expression changes grouped into GO processes including regulation of mitochondrial organization, apoptotic mitochondrial changes, mitochondrial fission and cristae organization, mitochondrial membrane permeability, and respiratory chain assembly. Given these findings, the group then undertook a comprehensive, high-resolution analysis of mitochondrial size, shape, and organization during HSV-1 infection. They used an arsenal of cutting-edge microscopy approaches including cryo soft X-ray tomography (SXT), serial block face SEM (SBF-SEM), ten-fold expansion microscopy (TREx) coupled with proximity ligation assay (PLA), and focused ion beam SEM (FIB-SEM) to demonstrate that HSV-1 infection causes mitochondria to become elongated, thin, rough, and more perinuclear, with more extensive ER-mitochondrial contact sites as well as shorter, thicker cristae. Consequently, they then analyzed mitochondrial function (oxygen consumption) during infection. Using Seahorse analytics they showed that while basal respiration and ATP production wane early in infection they are restored later, while mitochondrial proton leakage into the matrix and calcium uptake were associated with late infection. As a package this study serves to augment our understanding of transcriptional control and mitochondrial dynamics (structural and functional) during HSV-1 infection, and will likely be of broader interest to the host-pathogen community studying mechanisms of subversion of host energetics.

The manuscript is well-written, and appropriately referenced. The figures are generally clear and overall the data supports the authors’ conclusions. The balance between integral and supplemental figures/movies is appropriate, and the interactive transcriptomic Suppl. Table 1 is useful for data mining. (I was not able to log into the GEO dataset with the provided access token to examine that.) The supplemental movies are very helpful and impressive.

Reviewer #2: This study provides an in-depth examination of the effects of HSV-1 infection on mitochondria, offering key insights into the interactions between the virus and host cells. The findings indicate that HSV-1 infection leads to significant alterations in mitochondrial structure and function, including changes in mitochondrial morphology, thickening and shortening of cristae, an increase in the number and area of contact sites between mitochondria and the endoplasmic reticulum, as well as a rise in mitochondrial calcium ion content and proton leak. Although HSV-1 infection has been widely reported to affect the mitochondrial function of host cells, this work might be the first to provide direct morphological evidence at the microscopic level. These findings are interesting, and this manuscript suited to the wider audience of PLOS Pathogens. However, despite the significance of this research, the continuity of evidence in the article is insufficient, leading to certain logical leaps. Therefore, the major revisions are necessary before considering for publishing.

Reviewer #3: In this work, Leclerc et al attempted to demonstrate that infection with HSV-1 leads to significant transcriptional modification of host genes encoding proteins involved in the mitochondrial network, such as the respiratory chain, apoptosis and the structural organization of mitochondria. Infections carried out at different time points (2 hours and 8 hours) appeared to have an impact on the mitochondrial network.

The authors used high-resolution microscopy followed by interaction analysis to show that the mitochondrial network relocates to the perinuclear zone. In addition, the shape and structure of the mitochondria underwent morphological changes.

The authors also reported a significant increase in the number and clustering of mitochondrial ER contacts, along with a thickening and shortening of mitochondrial cristae. Metabolic analysis using Seahorse technology showed that ATP production was accompanied by an increase in mitochondrial Ca2+ content and a proton leak correlated with the stage of infection.

**Part II – Major Issues: Key Experiments Required for Acceptance**

Reviewer #1: It is puzzling that the authors selected the cell types they did for these studies, and I feel that these choices undermine the impact of the report.

MEFs and microscopy: Understandably, MEFs are amenable to microscopic approaches given their morphology. However, MEFs aren’t naturally infected with HSV-1, and their true value as an infection model (interrogation of gene deletions on infection) is not leveraged in this study. It is difficult to understand why in the very least a human cell line such as U2OS, hTERT-immortalized normal dermal fibroblasts, or better, a cell model for human neurons such as SK-N-SH, or SH-SY5Y, or best, iPSC-derived human neurons was not used for the mitochondrial morphology/structure analyses. Since mitochondrial organization and ultrastructure vary widely, using a more appropriate cell type would have elevated the impact of this study substantially. At the end of the study the reader interested in mitochondrial dynamics during a more natural HSV-1 infection is left asking “what about in human neurons?”, a question that will have to be answered by another study. Since the elegant microscopy approaches used here are highly specialized, requiring advanced technical expertise and beamline access, recapitulating these studies in neurons is not trivial for the majority of the interested readership.

Veros and transcriptomics, PLA: What is the rationale behind using Vero cells? The GRO-seq analysis of mitochondrial gene expression was carried out using HSV-1-infected Vero cells. Veros are instrumental in growing viruses owing in part to their lack of interferon production (Emeny and Morgan, J Gen. Virol., 43(1):247, 1979). Given this, using Veros for gene expression profiling at 4 and 8hpi, during which interferon responses are at their maxima in interferon-producing cells, will miss a plethora of interferon stimulated genes, many of which influence mitochondrial-associated innate immunity, activity, and shape. Mitochondrial antiviral defenses are mentioned regarding Figure 1A and B (Results, page 5, and again on page 8 before imaging section), but the transcriptional response shown lacks a large complement of ISGs.

The TREx/PLA and transcriptomic studies were also performed using Vero cells. Since the PLA is an extension of the work on mitochondrial-ER contact sites in MEFs, why was the cell type switched? What is the extent to which the MEF studies can be augmented by intramolecular interaction studies in a heterotypic cell line?

Reviewer #2: 1. The rationale for selecting the specific time points of 4, 8, and 12 hours post-infection (hpi) needs clarification. In some experiments, only 2 out of 3 time points were selected. It is recommended to supplement the missing time point data or should explain the reason.

2. In Fig 2j, there are only 4 cells at 8 hpi, significantly fewer than NI and 12 hpi groups. Is there a factor of artificial exclusion? What impact might this have on the statistical results?

3. In this work, the Mitochondrial permeability transition pore (mPTP) of Ca2+ entering mitochondria has not been given attention, especially the open state of this channel complex, would the author add this experiments?

In addition, the effect of HSV-1 virus protein on mPTP has been reported (PMID: 34234233, 34234233), and it is recommended to supplement the relevant discussion.

4. What is the basis for selecting VAPB and RNDM3 as representative proteins in the study of interactions between mitochondria and their inner membrane proteins?

5. Electron microscopy and 3D reconstruction are a major methodological innovation in this work, providing direct evidence of the impact of HSV-1 on host cell mitochondria. However, could the author provide direct evidence of the interaction between HSV-1 and mitochondria through SEM supplementation?

6. The imaging data related to mitochondria in the article is manually segmented; therefore, the basis for image segmentation should be described in the methods section. Furthermore, it seems that mitochondria cannot be visually distinguished from the image in Fig 2a, f.

Reviewer #3: Although the authors have presented original approaches with clear and demonstrative high-resolution images, the work presented in this article does not represent a major advance in the field. Indeed, there is no data specifying the potential role of HSV-1 during infection. No experiments specifying the impact of the virus on the perturbation of the mitochondrial network were presented. Important controls are missing, such as infection efficiency and viral replication. There is no imaging of colocalization between viral and mitochondrial proteins. These data could ultimately be presented with any pathogen/virus as we do not observe virus specificity during disruption of the mitochondrial network.

The seahorse experiments on mitochondrial metabolism have already been reported by other groups, as well as images of mitochondrial remodeling. It would be interesting to take this work further by highlighting the impact of the virus. Which viral proteins target mitochondria? Are there interactions between viral proteins and mitochondrial proteins or ER proteins? Is this phenomenon of mitochondrial disruption directly influenced by the presence of HSV-1 or is it an indirect phenomenon? This is interesting work, but highly incomplete.

**Part III – Minor Issues: Editorial and Data Presentation Modifications**

Reviewer #1: 1. General: The broad readership of PLOS Pathogens would very likely benefit from a table (supplemental) summarizing the different microscopy modalities used herein, ie: where that approach excels; what its basis is; what its benefits/drawbacks are (resolution vs. volumetric rendering), etc. Such a table would endow the reader with appreciation for the technical possibilities in ultrastructural biology (given beamline access), clarify the approach rationale, and make the figures easier to digest.

2. Font size and figure resolution: The font size and/or figure resolution in several areas needs to be increased. Figure 1 A and B should be as large as allowable so that they are readable. The node labels in 1A are unreadable even upon zooming into the PDF. In Figure 2, graphs C-J are very difficult to read, as are Figure 6 graphs B-G. Figure 6F would benefit also from even a modest enlargement. The font should be increased in Supplemental Figures 1A and 2B and C.

3. Page 3: Wording/clarity: “…complexes between mitochondria and the ER have been identified, including VAPB and PTPI51, respectively.”

4. Page 5: Wording/interpretation: The last sentence of the introduction is overly ge

---

## [Editor Report · Decision Letter 1]

12 Mar 2024

Dear Adjunct professor Vihinen-Ranta,

We are pleased to inform you that your manuscript 'Progression of herpesvirus infection remodels mitochondrial organization and metabolism' has been provisionally accepted for publication in PLOS Pathogens.

Best regards,

Eain A Murphy, Ph.D.

Academic Editor

PLOS Pathogens

Patrick Hearing

Section Editor

PLOS Pathogens

Michael Malim

Editor-in-Chief

PLOS Pathogens

orcid.org/0000-0002-7699-2064

Dr. Vihinen-Ranta,

Thank you for your resubmission. It is the opinion of the editors that the revised manuscript has been sufficiently improved and that it is now acceptable for publication in PLoS Pathogens.

Congratulations,

Eain Murphy
---

## [Editor Report · Acceptance letter]

5 Apr 2024

Dear Adjunct professor Vihinen-Ranta,

We are delighted to inform you that your manuscript, "Progression of herpesvirus infection remodels mitochondrial organization and metabolism," has been formally accepted for publication in PLOS Pathogens.

Best regards,

Michael Malim

Editor-in-Chief

PLOS Pathogens

orcid.org/0000-0002-7699-2064